# Harmonizing material quantity and terahertz wave interference shielding efficiency with metallic borophene nanosheets

Haojian Lin [1,2], Ximiao Wang[1,2], Hongjia Zhu [1,2], Zhaolong Cao[1], Jiahao Wu[1], Runze Zhan[1], Ningsheng Xu[1], Shaozhi Deng[1] ✉, Huanjun Chen [1] ✉ & Fei Liu [1] ✉

Materials with electromagnetic interference (EMI) shielding in the terahertz (THz) regime, while minimizing the quantity used, are highly demanded for future information communication, healthcare, and mineral resource exploration applications. Currently, there is often a trade-off between the amount of material used and the absolute EMI shielding effectiveness ($EES_t$) for the EMI shielding materials. Here, we address this trade-off by harnessing the unique properties of two-dimensional (2D) $\beta_{12}$-borophene ($\beta_{12}$-Br) nanosheets. By leveraging the high electron mobility and low mass density of $\beta_{12}$-Br, we simultaneously achieve a THz EMI shield effectiveness (SE) of 70 dB and an $EES_t$ of $4.8 \times 10^5$ dB·cm²/g (@0.87 THz) using a $\beta_{12}$-Br polymer composite. This surpasses the values of previously reported THz shielding materials with an $EES_t$ less than $3 \times 10^5$ dB·cm²/g and a SE smaller than 60 dB, while only needs 0.1 wt.% of these materials to realize the same SE value. Furthermore, by capitalizing on the superior mechanical properties of the composite, with 158% tensile strain at a Young's modulus of 21 MPa, we demonstrate the high-efficiency shielding performances of conformably coated surfaces based on $\beta_{12}$-Br nanosheets, suggesting their great potential in EMI shielding area.

THz radiation, spanning from 0.1 to 10 THz in frequency, represents a technological frontier with vast potential across applications such as information communications, radar imaging, noninvasive inspection, and biomedical sensing[1]. Its appeal lies in distinctive characteristics, including low photon energy, robust penetration capabilities, and the ability to discern molecular signatures within this spectral range. However, THz electronic and optoelectronic devices are susceptible to electromagnetic interference (EMI), a challenge well-documented in the radio-frequency domain[2–4]. Such interference can severely degrade device performance and harm the surrounding environment, underscoring the urgent need for effective EMI shielding materials.

An ideal EMI shielding material must effectively block electromagnetic waves (EMW) while ensuring that the isolated electromagnetic field does not radiate into the environment. Additionally, such materials should be lightweight, low-cost, easily producible, and possess good stretchability to conform to arbitrary surfaces. These properties are particularly critical for applications in wearable devices and portable consumer electronics, where flexibility and conformability are essential.

Current research on THz EMI shielding materials follows two primary approaches. The first relies on traditional materials designed for microwave frequencies, such as metal thin films[5–7] and carbon-based composites, including conductive carbon inks[8,9], carbon fibers[9,10], carbon nanotubes[9,11], and graphene[9,12]. While these materials are highly conductive and inhibit EMW through reflection, a significant portion of the waves is radiated into the environment, causing secondary pollution[5,13]. This limitation makes them unsuitable for miniaturized, high-integration devices. The second approach focuses on

[1]State Key Laboratory of Optoelectronic Materials and Technologies, Guangdong Province Key Laboratory of Display Material and Technology, and School of Electronics and Information Technology, Sun Yat-sen University, Guangzhou 510275, China. [2]These authors contributed equally: Haojian Lin, Ximiao Wang, Hongjia Zhu. ✉e-mail: stsdsz@mail.sysu.edu.cn; chenhj8@mail.sysu.edu.cn; liufei@mail.sysu.edu.cn

materials that shield EMW through absorption, leveraging free charged carriers within the material. Recent efforts have explored polymer composites[14,15], nanocomposites[16,17], multilayer assemblies[5,15], and porous foams/aerogels[17,18], which incorporate segregated conductive fillers characterized by a high density of free electrons and high electron mobility to facilitate EMW absorption. For instance, MXene assemblies have demonstrated near-intrinsic absorption limits in the 0.5–10 THz range, achieving this with an ultrathin thickness of just 10.2 nm[19]. Similarly, graphene/PMMA nanolaminates have reached shielding effectiveness (SE) values of up to 60 dB at a thickness of 33 μm[15]. In our recent study, we further showed that a transparent MXene film can achieve an SE of 21 dB across a broad frequency range of 0.1–10 THz[20].

Despite significant advances in the field, a critical issue remains: the trade-off between material mass and shielding performance. Specifically, current THz shielding materials exhibit $EES_t$ values below $3 \times 10^5$ dB·cm²/g and SE values under 60 dB (Supplementary Table 1)[15]. Improving performance often necessitates increasing the filler content or material thickness, which in turn compromises mechanical stability and restricts applicability in lightweight, flexible devices. Moreover, for practical deployment, there is an urgent need for environmentally sustainable and scalable manufacturing processes for these materials— yet this remains a significant challenge[2,21]. These persistent difficulties highlight the pressing demand for new materials that can combine high $EES_t$ and SE values with flexibility, low density, and scalable production.

Borophene, the boron analog of graphene, emerges as a promising candidate to meet these demands. As the lightest metalloid[22], boron imparts borophene with ultralow density, rendering it inherently lightweight—a critical advantage for applications in portable and wearable electronics. Moreover, borophene exhibits a high carrier density ($3.3 \times 10^{19} \sim 4.3 \times 10^{19}$ m⁻²)[23] and exceptional electron mobility ($878.6 \sim 28.4 \times 10^5$ cm²V⁻¹ s⁻¹)[24,25], properties essential for achieving strong THz absorption while minimizing material usage. These characteristics enable borophene to effectively dissipate EMW through scattering and re-absorption by free electrons, significantly reducing the quantity of material required to achieve high shielding performance. In addition to its electronic properties, structural versatility of borophene[22,26]—manifested in multiple phases such as triangular, honeycomb, stripe, and rectangular configurations—endows it with exceptional flexibility and mechanical strength, as evidenced by its high Young's modulus (170 ~ 398 GPa·nm)[26]. This flexibility makes borophene an ideal filler for polymer matrices, enabling it to conform to arbitrary surfaces and meet the demands of advanced EMI shielding applications in flexible electronics. By leveraging the unique combination of ultralow density, high carrier density, and outstanding electron mobility in borophene, it is possible to overcome the trade-off between material quantity and THz EMI performance. This positions borophene as a transformative material for next-generation EMI shielding, offering a pathway to achieve high SE with minimal material usage while retaining the flexibility required for practical applications. However, despite these advantages, the development of borophene-based EMI shielding materials has been severely hindered by the lack of a scalable synthesis method for high-yield, freestanding metallic borophene. As a result, reported $EES_t$ values remain below 0.125 dB·cm²/g, and SE values are lower than 42 dB[27,28]—far from realizing its full potential.

In this study, we address the above limitations by developing a scalable synthesis method for wafer-scale polydimethylsiloxane (PDMS) composites embedded with few-layer $\beta_{12}$-Br single-crystalline nanosheets, which exhibit extraordinary free electron density and excellent electrical conductivity ($3.0 \times 10^4$ S/m). Due to strong THz resonances initiated by the free electrons, the composite achieves a mean SE of over 70 dB and an unprecedented $EES_t$ of $4.8 \times 10^5$ dB·cm²/g —the highest reported value to date—across the 0.5 to 2 THz range,

with potential extension up to 10 THz. These results surpass those of previously reported THz EMI shielding materials, while utilizing only 1/10000th to 1/100th of the mass required by other materials (Supplementary Table 1). Furthermore, the composite exhibits remarkable flexibility, with a tensile strain exceeding 158% at a tensile stress of 33 MPa, enabling effective shielding on conformably coated surfaces. Based on the $\beta_{12}$-Br/PDMS composites, we further demonstrate their practical applications in EMW shielding of objects with irregular surfaces. This work not only demonstrates the potential of few-layer metallic $\beta_{12}$-Br nanosheets as highly efficient, low-density, and elastic materials for THz shielding but also addresses critical challenges in the field. By providing a scalable synthesis method and achieving record-breaking performance, we pave the way for borophene-based composites in next-generation EMI shielding applications, particularly in portable, wearable, and miniaturized devices where lightweight, flexibility, and high performance are paramount.

## Results

### Fabrication and characterization of $\beta_{12}$-Br/PDMS composites
The $\beta_{12}$-Br comprises five atoms per unit cell, characterized by alternating rows of empty and filled hexagons along the x-direction. This arrangement usually produces stripes of vacancies along this axis. Along the y-direction, the structure consists of columns featuring a continuous line of atoms interspersed with incomplete hexagons, as shown in Fig. 1a. The unique crystalline structure of $\beta_{12}$-Br results in a high electron density ($3.4 \times 10^{19}$/m²), accompanied by a large electron mobility ($878.6 \sim 2.84 \times 10^6$ cm²/(V·s)) and a high Young's modulus (170 ~ 398 GPa·nm)[23–26]. Furthermore, it is the most stable among the various allotropes of borophene in its freestanding state[26,29–32]. In this study, the borophene nanosheets were synthesized by our developed low-temperature liquid phase exfoliation (LTLE) technique (see "Methods" for details)[33], as depicted in Fig. 1b. The crystalline structure and stoichiometric ratio of the products are respectively confirmed by X-ray diffraction (XRD) and confocal Raman spectroscopy, revealing the nanosheets are the $\beta_{12}$ phase with high crystallinity (Supplementary Figs. 1 and 2). The $\beta_{12}$-Br nanosheets are a few atomic layers in thickness and well-dispersed into an aqueous solution, with an average thickness about 4 nm (Fig. 1c, and Supplementary Fig. 3a) and a six-fold symmetry (Fig. 1d, and Supplementary Fig. 4). Additionally, over 94 at.% of the boron atoms remain unoxidized (Supplementary Figs. 5 and 6).

It should be noted that fabricating robust, stretchable, and continuous thin film based on pure borophene nanosheets alone has significant challenges, especially when attempting to scale up to larger area[27]. Instead, we employ the $\beta_{12}$-Br nanosheets as the fillers embedded into PDMS film to form a flexible composite. Specifically, the composite film was produced using a simple sol-gel method (Fig. 1b), where the $\beta_{12}$-Br nanosheets were intricately linked together via robust intermolecular bonding interactions involving hydrogen atoms from PDMS molecules and boron atoms located on the surface of the $\beta_{12}$-Br (Fig. 1a). This approach has been previously employed, where α-Br flakes were used as fillers[27]. However, due to the inefficient EMI shielding of α-Br reported in those studies, elevated concentration of borophene, up to 100 wt.%, was necessary. This undoubtedly causes the difficulties in achieving uniform and continuous thin films owe to potential cracking of the PDMS matrix. For the $\beta_{12}$-Br, as discussed below, leveraging the ultrahigh $EES_t$ of the $\beta_{12}$-Br allows for exceptionally low borophene content of less than 0.5 wt.% in the matrix and minimizing the breakage probability of PDMS intermolecular bond. This obvious decrease of the $\beta_{12}$-Br content facilitates their homogeneous dispersion within the PDMS matrix (Fig. 1f, inset), enabling the formation of flexible large-area thin films up to 5 inches (Fig. 1e, f).

In-situ measurements are conducted on individual $\beta_{12}$-Br nanosheet to explore its native electrical transport behaviors, as shown in Fig. 2a. Based on the I − V characteristics exhibit a clear linear

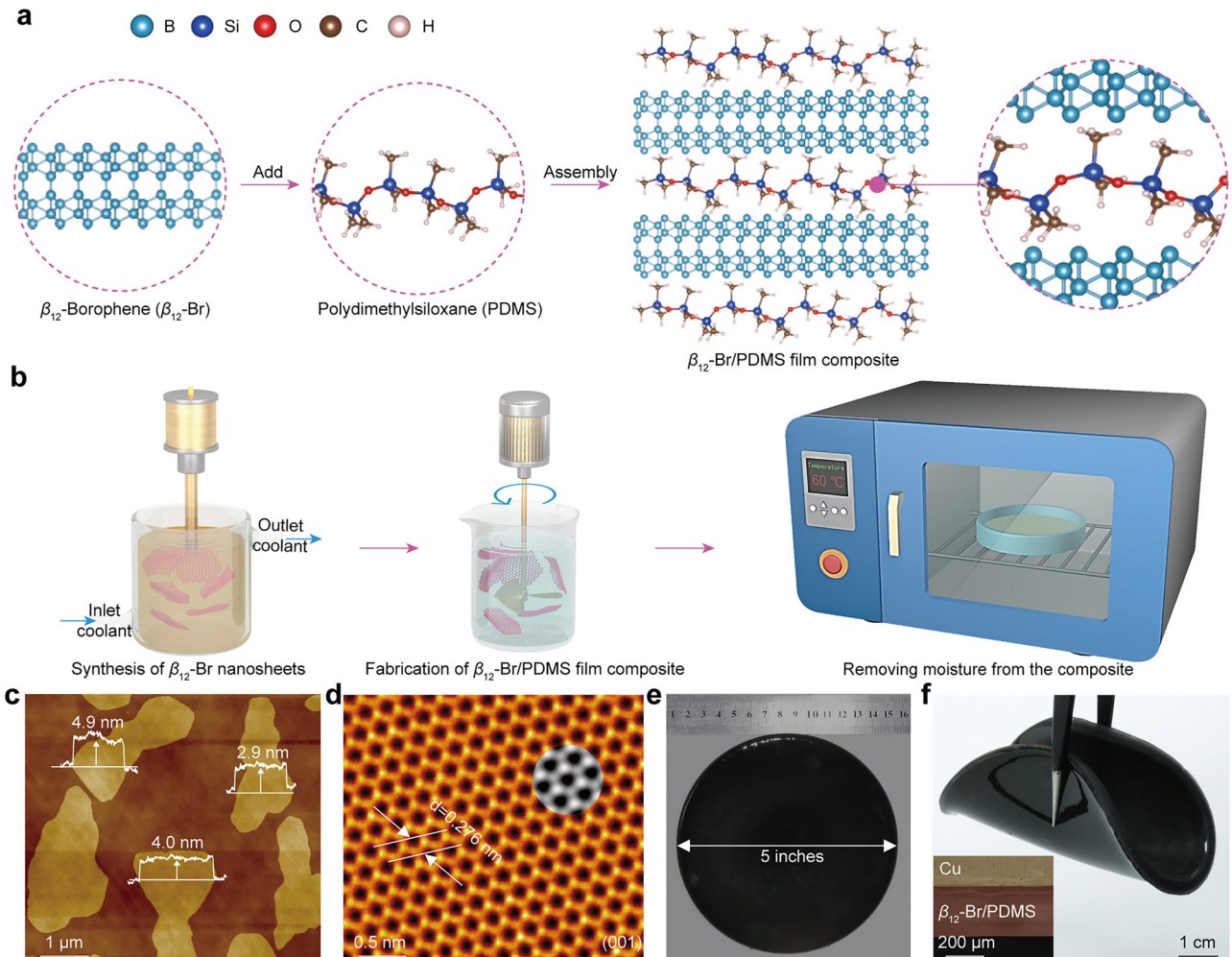

**Fig. 1 | Characterization of $\beta_{12}$-Br/PDMS composite film. a** A schematic diagram illustrating the preparation process of $\beta_{12}$-Br nanosheets and $\beta_{12}$-Br/PDMS composite film. **b** The production route for $\beta_{12}$-Br/PDMS composite film. **c** An AFM image displaying the topography of few-layer $\beta_{12}$-Br nanosheets synthesized by LTLE way, where the monolayer thickness is indicated by the inserted height profile. **d** A typical HRTEM image of a $\beta_{12}$-Br nanosheet, with the inset presenting the theoretical surface configuration of borophene (001) face using density functional theory. **e** Representative photograph showcasing the as-synthesized centimeter-scale $\beta_{12}$-Br/PDMS composite film. **f** Photograph demonstrating the excellent flexibility of the composite film, with an inset showing the corresponding cross-section image. The $\beta_{12}$-Br thickness data generated in this study are provided in the Source Data file.

relationship (inset), the average electrical conductivity of individual nanosheets can be calculated to be about $3.0 \times 10^4$ S/m, suggesting their metallic conduction behaviors. It is noted that the electrical conductivity of a single $\beta_{12}$-Br nanosheet with a 4-nm thickness is two orders of magnitude higher than that of individual $MoS_2$ (303.03 S/m), $WS_2$ (416.67 S/m), and black phosphorus (6400 S/m)[34,35]. Figure 2c gives the morphology and measurement circuit of bare $\beta_{12}$-Br nanosheet film with a 400-nm thickness. As presented in Fig. 2d, the mean sheet resistance of the $\beta_{12}$-Br nanosheet film is determined to be $1.7 \times 10^4$ $\Omega$/sq according to the I−V curves (inset), which is approximately five orders of magnitude lower than that ($6.0 \times 10^9$ $\Omega$/sq) of the nanostructure film consisted of the mixed $\beta_{12}$-Br and $\chi_3$-Br phases[36,37]. The above transport results prove the superior crystallinity and purity of $\beta_{12}$-Br sheets synthesized by our method, suggesting the $\beta_{12}$-Br nanosheets with high electrical conductivity hold great potential for THz-wave EMI shielding applications.

### THz shielding performance and mechanism of $\beta_{12}$-Br/PDMS composite film

The THz-wave shielding performance of the $\beta_{12}$-Br/PDMS composite film was evaluated using a THz time-domain spectroscope (THz-TDS) system, as given in the inset of Fig. 3a. From Fig. 3a, the excellent EMW

shielding performances of 0.6-mm-thick composite film can extend across a very wide frequency range from 0.1 to 7 THz, highlighting that the $\beta_{12}$-Br/PDMS composite film serves as an ultra-broadband EMW shielding material. To better comprehend the contribution of different components to the THz shielding performance of the composite film, both the reflection and transmission spectra of the composite film are measured together to obtain the EMI absorption effectiveness ($SE_A$) and reflection effectiveness ($SE_R$), respectively. It is obviously seen that the average $SE_A$ reaches up to 60 dB while the $SE_R$ is only ~5 dB when the illumination frequency is larger than 0.2 THz, revealing that the absorption loss should dominate over the THz-wave shielding behaviors of $\beta_{12}$-Br/PDMS composite film rather than the reflection or transmission loss. With a thickness of 2 mm and containing 0.5 wt.% of $\beta_{12}$-Br, the EMI SE of the composite film can reach as high as 70 dB (Fig. 3c). Furthermore, the EMI $EES_t$, which offers a more comprehensive assessment of the shielding performance of the material under idealized conditions by considering the mass of the shielding material and the spot size of incident EMW (see "Methods"), is obtained to be in the range of $2.5 \times 10^5 - 4.8 \times 10^5$ dB·cm$^2$/g across the frequency range of 0.8 THz to 2 THz. Notably, several films, comprising of graphene or MXene composites among others, demonstrate the EMI $EES_t$ maxima capped at $3 \times 10^5$ dB·cm$^2$/g. However, these composite materials, even

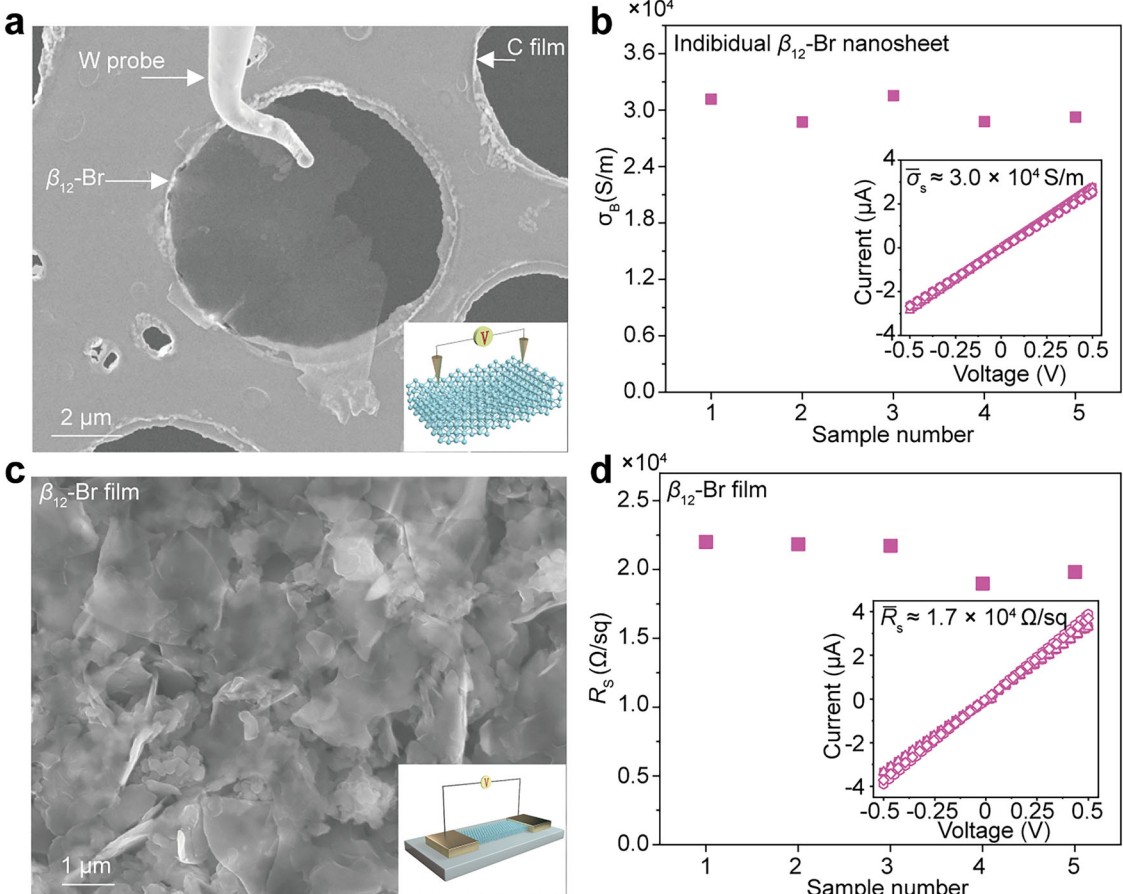

**Fig. 2 | Electrical transport properties of $\beta_{12}$-Br nanosheet. a** SEM image of an individual $\beta_{12}$-Br nanosheet during in-situ measurements, and the corresponding schematic diagram is shown in the inset. **b** Electrical conductivity distribution of a single $\beta_{12}$–Br nanosheet. Their corresponding I–V characteristics are shown in the inset. **c** SEM image of a piece of $\beta_{12}$-Br nanosheet film laid on interdigital electrodes. The inset gives the measurement circuit of sheet resistance. **d** Sheet resistance of a bare $\beta_{12}$-Br nanosheet film, and the inset gives their representative I-V curves. The data for I-V curves generated in this study are provided in the Source Data file.

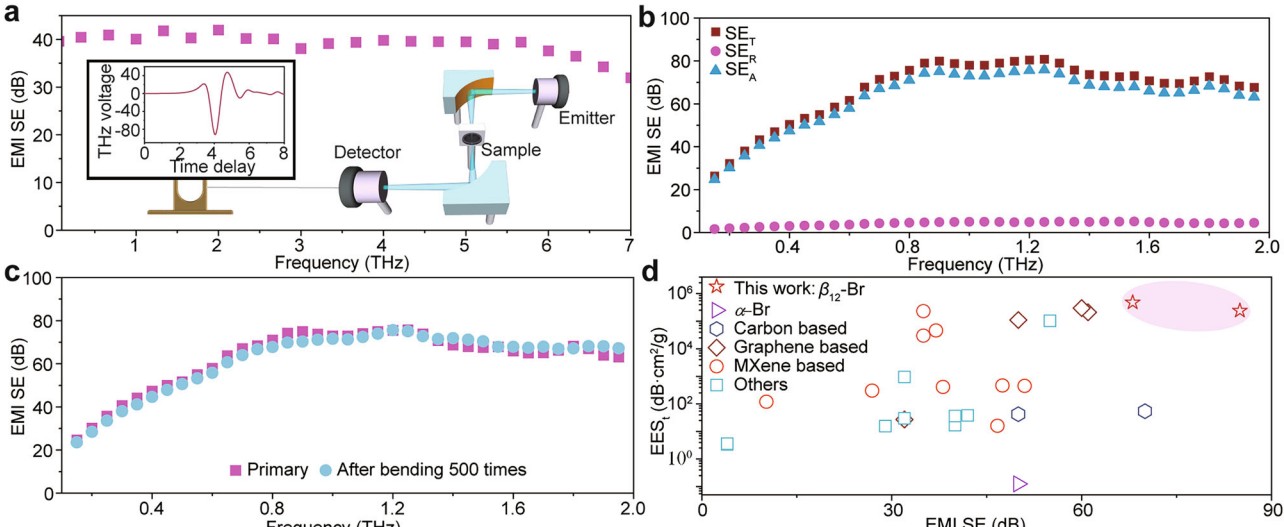

**Fig. 3 | THz shielding measurements of $\beta_{12}$-Br/PDMS composite film. a** The EMI SE curve of 0.6-mm-thickness 0.5 wt.% $\beta_{12}$-Br/PDMS composite film in an ultrawide frequency range of 0.1 – 7 THz. The inset schematically illustrates the optical path for THz-TDS measurements. **b** The curves of transmission, reflection, and absorption shielding efficiencies of 0.5 wt.% $\beta_{12}$-Br/PDMS composite film with a 2-mm-thickness versus THz frequency, respectively. **c** The frequency-dependent analysis of EMI SE curves of the 2-mm-thickness composite film with a 0.5 wt.% $\beta_{12}$-Br nanosheets before and after undergoing 500 bending cycles. **d** Comparison between EMI SE and EMI EES$_t$ values obtained from $\beta_{12}$-Br/PDMS film and other excellent shielding materials of $\alpha$-Br[27], carbon-based[8,47], graphene-based[12,13,15,16,48,49], MXene-based[18,50–52], and others[14,16,53–60]. The EMI SE and EES$_t$ data generated in this study are provided in the Source Data file.

at concentrations of at least 5 wt.%, fell short of achieving the benchmark of 80 dB in terms of EMI SE (Fig. 3d, and Supplementary Table 1). In stark contrast, the $\beta_{12}$-Br/PDMS film, comprising a mere 0.13 wt.% weight fraction and measuring 2 mm in thickness, showcased an impressive average EMI EES$_t$ of $4.8 \times 10^5$ dB·cm$^2$/g. Furthermore, with an increase in the weight fraction to 0.5 wt.% and the thickness to 4 mm, the film attained an average EMI SE as high as 85 dB. Such comparison evidently underscores that despite of the evidently improved performance of the $\beta_{12}$-Br/PDMS film, the mass of the shielding materials is four orders of magnitude smaller than that of other excellent shielding materials previously reported.

A series of $\beta_{12}$-Br/PDMS composite films were prepared to investigate the influence of the weight fraction of $\beta_{12}$-Br nanosheets and the film thickness on the THz EMI shielding performance (Supplementary Fig. 7). The power of THz EMW is effectively dissipated by the composite film, specifically by the $\beta_{12}$-Br nanosheets, which is elucidated by monitoring the SE spectra measured from composite films with varying concentrations of $\beta_{12}$-Br nanosheet but identical thickness (2 mm). By progressively increasing the borophene content from 0.13 to 2.1 wt.%, the EMI SE initially experiences rapid augmentation, eventually reaching a plateau for weight fraction over 0.5 wt.%. Notably, the highest SE achieved can arrive at 76 dB at 0.87 THz for a weight fraction of 2.1 wt.%, unveiling that the EMW shielding of the composite film is primarily attributed to the addition of $\beta_{12}$-Br nanosheets.

The aforementioned results clearly show that even a small (2.1 wt.%) addition of $\beta_{12}$-Br nanosheet into the polymer matrix can yield a superhigh EMI SE of 76 dB. This obviously surpasses previous findings, where an SE of only 42 dB was achieved with 100 wt.% of $\alpha$-Br filled into the same matrix[27]. Most of all, for a weight fraction of 0.13 wt.%, the EMI EES$_t$ can reach up to $4.8 \times 10^5$ dB·cm$^2$/g, superior to all reported composite films with conductive fillers such as graphene and MXene (Fig. 3d). Simultaneously, the EMI SE value of $\beta_{12}$-Br/PDMS composite film can still maintain as high as 68 dB, which achieves the best value reported up to date.

The addition of $\beta_{12}$-Br nanosheet fillers is found to be very essential for evidently improving the THz shielding performance of composite film. If the filler is changed from bulk boron powers to the $\beta_{12}$-Br nanosheets, the THz shielding performances of the composite film were significantly improved (Fig. 4a). For instance, in a composite film with 0.13 wt.%, the SE already reaches as high as 68 dB at 0.87 THz. To probe the decisive factors of the EMW shielding performance of the $\beta_{12}$-Br/PDMS composite film, the dependence of EMI EES$_t$ on the weight fraction of borophene nanosheets is characterized, revealing a nonlinear increase with the reduction of the weight fraction (Fig. 4b). Specifically, by progressively increasing the borophene content from 0.13 to 2.1 wt.%, the EMI SE initially experiences rapid augmentation, eventually reaching a plateau for weight fraction over 0.5 wt.%. Such saturation effect can be attributed to the competition between absorption and scattering of incident THz wave by the $\beta_{12}$-Br nanosheets at different borophene content. According to previous studies[38,39], THz wave absorption by the $\beta_{12}$-Br nanosheets will first monotonically increase with the borophene content. When the borophene content arrives at a critical value, the maximum THz absorption of the $\beta_{12}$-Br nanosheets will be achieved. But if the borophene content is continuously increased to exceed the critical value, the scattering coefficient will sharply increase, originating from the strong scattering loss of the incident THz waves by the nanosheet assemblies inside the PDMS matrix instead of being fully absorbed by each $\beta_{12}$-Br nanosheet. This will lead to suppression of THz wave absorption and therefore saturation of the EMI SE (see more detail discussion in Supplementary Note 1 and Supplementary Fig. 8).

Another reason for the saturation behavior can be attributed to the detection limit of the THz-TDS system used in our study. According to the shielding efficiency equation:[5]

$$\text{EMI SE (dB)} = 20 \log_{10} \left( \frac{I_{in}}{I_{out}} \right) \qquad (1)$$

where $I_{in}$ an $I_{out}$ represent the intensities of the incident and transmitted THz waves, respectively. When the borophene content reaches ~0.5 wt.%, the THz EMI SE achieves a value of up to 80 dB. This corresponds to 99.99% of the incident wave being absorbed, with only 0.01% of the wave transmitting through the composite film. The detection limit of the THz-TDS spectrometer used in our measurements (BATOP, TDS 1008) is 85 dB. This means that when the transmitted wave intensity falls below $5 \times 10^{-3}$% of the incident wave, the spectrometer can no longer reliably distinguish intensity differences associated with varying transmission levels. As a result, the THz EMI SE of the $\beta_{12}$-Br/PDMS film appears to saturate in our experiments once the borophene content exceeds 0.5 wt.%, remaining nearly constant with further increases in borophene content.

The THz-wave shielding performance of the composite film is also highly dependent on its thickness, as evidenced by the EMI SE spectra plotted against the composite film thickness (Fig. 4c). If the weight fraction of borophene nanosheets was kept at 0.5 wt.%, both of the EMI SE and EES$_t$ values monotonically increase with the composite film thickness (Fig. 4d). Particularly, for a composite film thickness of 4 mm, the maximum EMI SE and EES$_t$ can reach as high as 85 dB and $2.5 \times 10^5$ dB·cm$^2$·g$^{-1}$ at 0.87 THz, respectively. These results evidently demonstrate the exceptional EMW shielding performance of the $\beta_{12}$-Br/PDMS composite film.

Afterwards, the mechanism governing the strong THz EMW absorption of the $\beta_{12}$-Br/PDMS composite film are further explored. The preceding discussion unambiguously suggests that the incident EMW are efficiently absorbed and dissipated by various $\beta_{12}$-Br nanosheets within the composite film. This can be attributed to the electronic and structural properties of $\beta_{12}$-Br nanosheets. Each single-crystalline $\beta_{12}$-Br nanosheet supports a remarkably high concentration of free electrons with exceptionally large mobility[23], which are among the highest reported values for 2D materials. When exposed to THz waves, these free electrons are excited and accelerated by the electric field of the EMW, resulting in collective electron oscillations within the nanosheets (Fig. 4e). The planar size of the $\beta_{12}$-Br nanosheets (~5 μm, as shown in Fig. 1c and Supplementary Fig. 3b) is significantly smaller than the wavelength of the incident THz radiation, causing the excited electrons to encounter the nanosheet boundaries and undergo multiple reflections. These reflections act as a restoring force, giving rise to strong resonances in the THz spectral region (Supplementary Figs. 10 and 11). Due to the subwavelength size of the nanosheets, most of these resonances decay via absorption rather than scattering (Supplementary Fig. 12), a phenomenon well-explained by the principles of electromagnetic absorption and scattering by small particles[40]. As a result, the kinetic energy of the oscillating electrons is efficiently transferred to the lattice of the $\beta_{12}$-Br nanosheets, converting nearly all of the absorbed EMW into heat. This energy dissipation process, driven by the high electron mobility, confinement-induced resonances, and dominant absorption mechanism, leads to the ultrahigh EMI absorption effectiveness observed in our study.

This mechanism can be further supported by simulating the dynamics of electrons and the corresponding absorption of EMW energy within a $\beta_{12}$-Br nanosheet (Methods and Supplementary Note 2). The behavior of free electrons in $\beta_{12}$-Br nanosheet in response to the EMW is governed by the alternating-current conductivity, which can be characterized using the Drude model (Supplementary Fig. 9). For a nanosheet with a monolayer thickness, electromagnetic resonance characterized by strong absorption peaks can be excited upon irradiation by the THz wave (Supplementary Fig. 9). These resonances induce strong electron oscillations within the nanosheet, which are

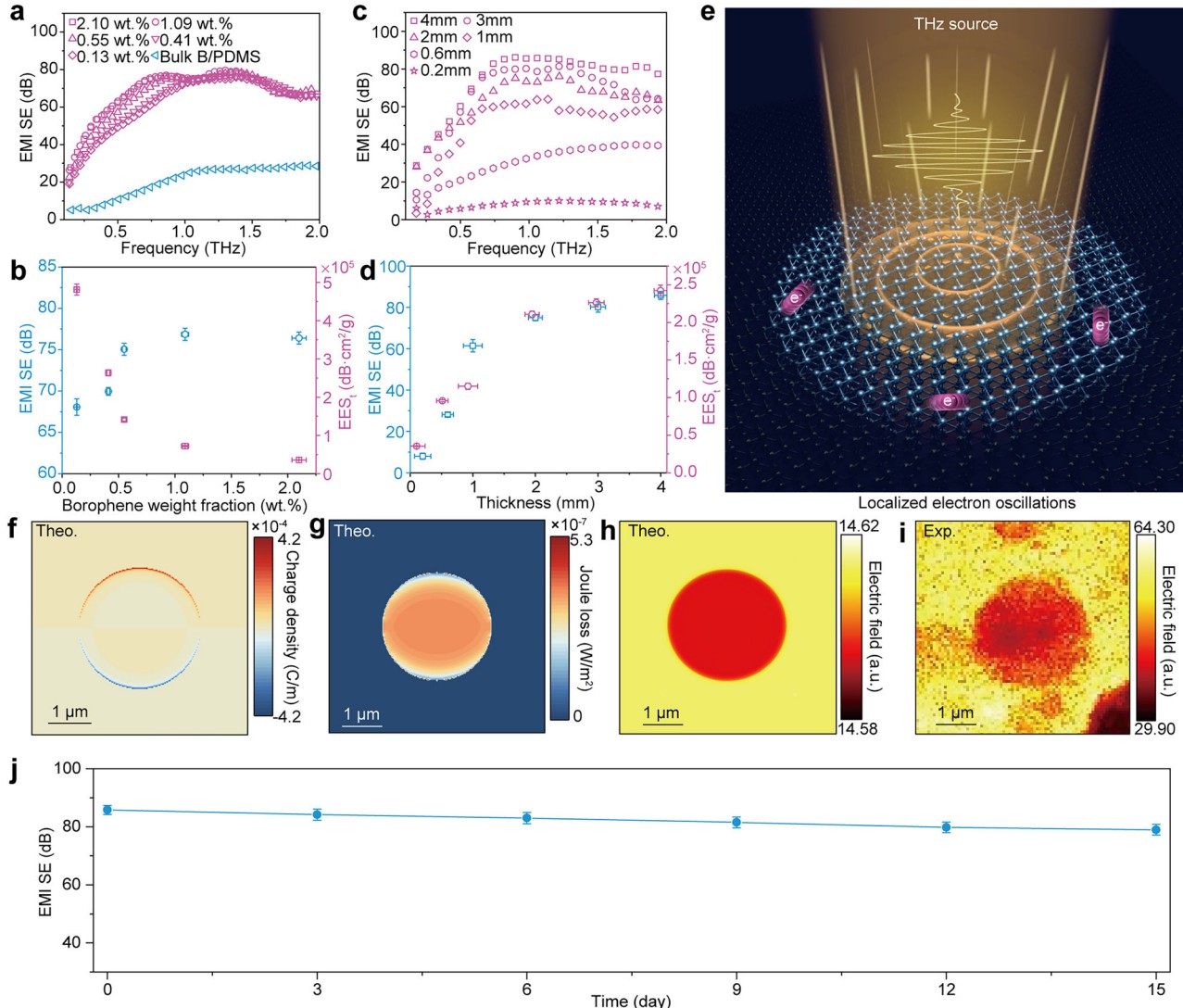

**Fig. 4 | EMI shielding properties of $\beta_{12}$-Br/PDMS composite film. a** The EMI SE curves of the composite film with various weight fractions of $\beta_{12}$-Br nanosheets. **b** The curves of EMI SE/EMI $EES_t$ to the weight fraction of borophene nanosheets at 0.87 THz. **c** The frequency-dependent EMI SE curves of the composite film with different thickness. **d** The relationship between EMI SE/EMI $EES_t$ and film thickness at 0.87 THz. **e** Schematic showing the excitation of electron oscillations in $\beta_{12}$-Br nanosheets under THz radiation. The orange annular ripple represents the collective oscillation of the electrons in $\beta_{12}$-Br nanosheet induced by the THz electric field, while the purple balls stand for every individual oscillating electron. **f, g** Simulated spatial distributions of charge density (**f**) and Joule loss (**g**) within an individual $\beta_{12}$-Br nanosheet. **h, i** Simulated (**h**) and experimental (**i**) near-field image of an individual $\beta_{12}$-Br nanosheet on a 500-µm-thick highly doped *n*-silicon substrate. The simulations and experimental measurements are conducted in the frequency range of 0.5 – 2 THz. The diameters of the circular nanosheets are all set at 5.64 µm, as determined by the experimental measurement shown in (**i**). **j** The EMI SE stability of the $\beta_{12}$-Br/PDMS composite film after 15-day air storage. Note: All error bars represent the standard deviation derived from three independent measurements. The EMI SE and simulations data generated in this study are provided in the Source Data file.

subsequently reflected by the nanosheet boundary (Fig. 4f). This results in significant Joule loss occurring within the borophene nanosheets (Fig. 4g), leading to attenuated electromagnetic fields inside the nanosheet compared to the substrate (Fig. 4h). This phenomenon is further corroborated by THz near-field optical measurements conducted on an individual $\beta_{12}$-Br nanosheet (Fig. 4i, see "Method" for numerical simulations). As a result, the Joule loss contributes to a remarkably high absorption efficiency of up to 43% for the nanosheet array at resonance, even for a monolayer thickness (Supplementary Fig. 9). This highlights the efficient energy dissipation mechanism enabled by the rich free electrons within the $\beta_{12}$-Br nanosheets.

It should be noted that the experimental THz nano-imaging was conducted on the samples using a scattering-type THz optical microscope (THz-NeaSNOM, Neaspec GmbH, see "Method" for details). The

nano-imaging system is equipped with a broadband THz operating in pulse mode, which covers a frequency range of 0.5 to 2.0 THz. Due to the limitations of this light source, we were only able to conduct broadband near-field imaging. As a result, Fig. 4i represents the outcome obtained through broadband acquisition of the near-field THz wave amplitude. In the simulations, to ensure consistency with the experimental conditions, we employed the same broadband source (0.5 to 2.0 THz) in our simulations to excite an individual borophene nanosheet and performed spectral integration of the electron concentration (Fig. 4f) and Joule loss (Fig. 4g) over the same spectral range. For the near-field THz wave amplitude, in the simulation the perpendicular component of the electric field, $|E_z|$, was extracted in the frequency domain (0.5 to 2.0 THz) at a plane 100 nm above the surface and converted back to time-domain near-field optical maps via inverse Fourier transform. The $|E_z|$ was then integrated over the time

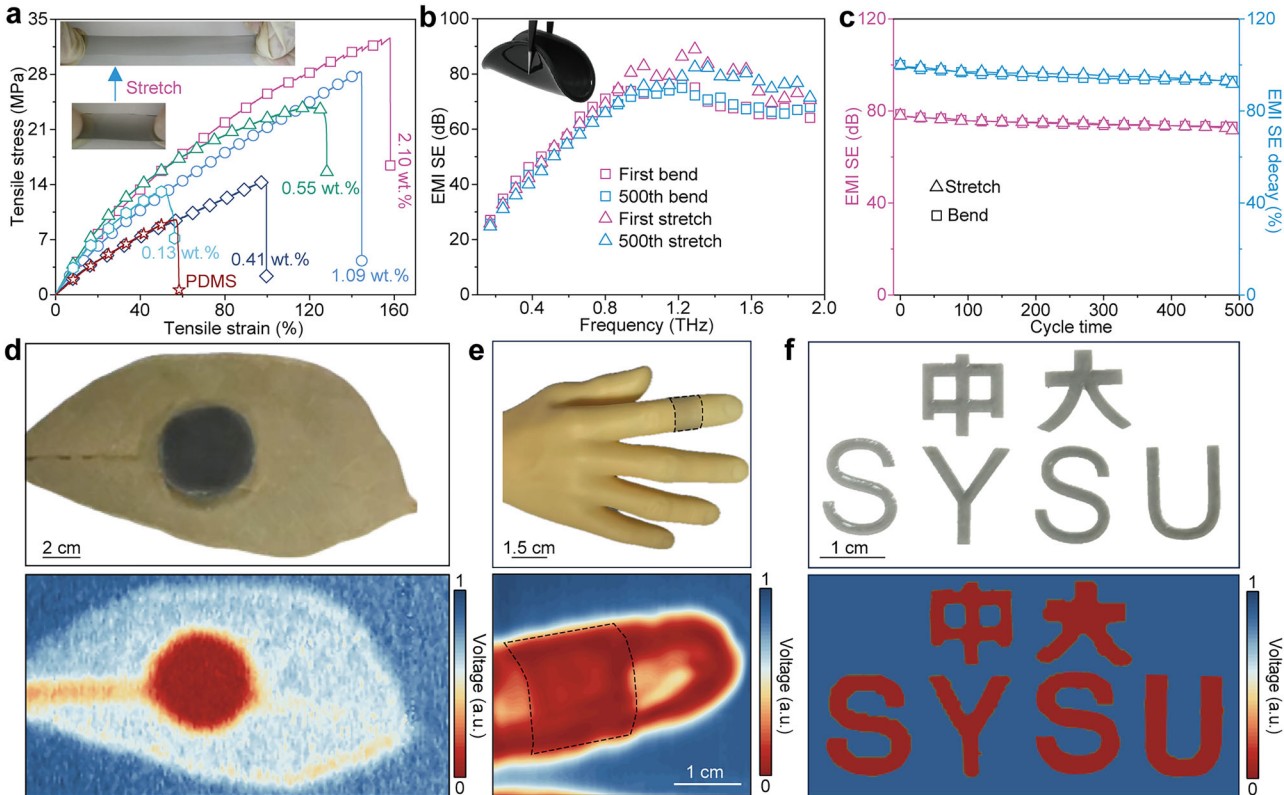

**Fig. 5 | Mechanical properties of $\beta_{12}$-Br/PDMS composite films and their THz shielding applications. a** Representative stress-strain curves of the 2-mm-thickness composite film with different borophene contents in uniaxial tensile test. The inset at the top left corner shows a photograph of the composite film undergoing tensile testing. **b** EMI SE values of the composite film after 500 times' bending or stretching measurements. The photograph of the composite film undergoing bending is displayed in the inset. **c** The EMI SE value and EMI SE decay ratio of the composite film at 0.87 THz during 500 bending or stretching cycles, respectively. **d, e** Digital images of the leaf coating and the human finger coated by the planar and curved composite film, respectively, in which the THz wave is effectively shielded and the inner information can't be differentiated. **f** THz imaging of the Sun Yat-sen University masked by shaped $\beta_{12}$-Br/PDMS composite film. The stress-strain and EMI SE data generated in this study are provided in the Source Data file.

domain to obtain the near-field distribution, as shown in Fig. 4h (see Supplementary Note 2 for more simulation details).

According to the simulation results, the resonance frequency of the $\beta_{12}$-Br nanosheet is strongly dependent on its size, shape, and thickness, covering a broad spectral range (Supplementary Fig. 10a and 11a). The maximum absorption efficiency remains nearly constant at 43% (Supplementary Fig. 10c), regardless of the shapes of the borophene nanosheets, while the lateral size of the nanosheet strongly influences the absorption cross-section of the individual nanosheet (Supplementary Fig. 10b). The THz absorption intensity is also influenced by the nanosheet thickness. Specifically, when the borophene thickness is ≤500 nm (Supplementary Fig. 11b), the absorption of the array stabilizes at 44%, which is comparable to that of a monolayer borophene nanosheet array (43%). As the nanosheet thickness increases further, a reduction in absorption intensity is observed. Given that the thicknesses of the borophene nanosheets used in our study is less than 5 nm (Fig. 1c and Supplementary Fig. 3a), the broadband yet relatively stable THz EMI absorption shielding performance observed in our experimental measurements for the $\beta_{12}$-Br/PDMS composite film can be attributed to the presence of nanosheets with varying sizes and shapes within the composite film.

Interestingly, even after being exposed to air over 15 days, the EMI shielding performances of the $\beta_{12}$-Br/PDMS composite film showed no obvious degradation and retained 92.94% of the original performance (Fig. 4j), revealing the high stability of the $\beta_{12}$-Br/PDMS film under ambient conditions.

## Mechanical properties of the $\beta_{12}$-Br/PDMS composite film

Good mechanical property is another critical requirement for THz EMW shielding materials in portable and wearable electronic devices[5,41]. By taking advantage of the exceptional Young's modulus of $\beta_{12}$-Br (theoretical value of 170 to 398 GPa·nm)[26], the composite film can be easily bent up to a high angle of 180° without any breakage or cracks observed through 500 bending experiments, revealing its excellent flexibility. Further uniaxial tensile experiments were conducted to quantitatively assess the elastic behavior of the composite film with various weight fractions of borophene nanosheets. It is noted that the 0.13 wt.% $\beta_{12}$-Br/PDMS composite film exhibits a tensile strain, $\sigma_s$, of 50% at a tensile stress, $\varepsilon_s$, of 13 MPa, while that with 2.10 wt.% achieves a $\sigma_s$ of 158% at $\varepsilon_s$ = 33 MPa (Fig. 5a). The corresponding Young's moduli, $E_c = \sigma_s/\varepsilon_s \times 100$[42], were ~26 MPa and 21 MPa, respectively. One can obviously see that the mechanical properties of the composite film are gradually enhanced with increasing the weight fraction of borophene, overwhelming those of many other high-modulus 2D materials (Supplementary Table 3). Also, this observed trend corroborates that the ultrahigh theoretical Young's modulus of borophene nanosheets should be responsible for the excellent mechanical properties of the composite film.

THz shielding measurements were then performed on the 2-mm-thick composite film with 0.5 wt.% $\beta_{12}$-Br nanosheets. After 500 bending cycles, the mean EMI SE value of the film slightly decreased from 81 dB to 75 dB, with an average fading rate as low as 0.015% per bending cycle (Fig. 5b). Similarly, the mean THz EMI SE value of the composite film mildly reduced from 78 dB to 71 dB after 500 stretching

cycles, with an average fading rate of less than 0.018% per stretching cycle (Fig. 5c). Despite of these tiny attenuation, the sample can recover over 80% of its initial EMI SE efficiency for THz waves after both bending and stretching tests. These findings validate the great potential of $\beta_{12}$-Br/PDMS composite film to combine high flexibility with superior mechanical strength, making them promising candidates for advanced flexible electronic applications.

### Application demonstration of the $\beta_{12}$-Br/PDMS composite film

The discussion above clearly exhibits the excellent THz shielding performance and flexibility of the $\beta_{12}$-Br/PDMS composite film, which ensure them highly suitable for applications in EMW shielding of objects with irregular surfaces. To this end, the $\beta_{12}$-Br/PDMS composite film was employed to conformally coated onto practical objects to show its excellent EMI SE upon THz EMW illumination. A 1-mm-thick $\beta_{12}$-Br/PDMS composite film was placed on the surface of a dry leaf for THz shielding imaging (Fig. 5d). As observed in Fig. 5e, the THz signal is almost completely absorbed where the composite film covers the leaf, causing the emergence of the shadow for imaging leaf. Additionally, the same composite film was wrapped around a human finger, effectively blocking off the THz signal at the wrapped region. Moreover, the composite film can also be fabricated into different patterns to match the outlines of target objects for demonstrating different characters, as presented in Fig. 5f. It is obviously seen that the contour profile of "Sun Yat-sen University" is very clear and sharp, further showcasing the versatility of the composite film in being processed into EMW shielding materials with arbitrary shapes to meet diverse application requirements.

## Discussion

Borophene, an important member of 2D material family, has attracted much attention since the birth due to its unique electrical and mechanical properties, as well as low mass density. While their applications in energy conversion and storage have been widely studied, their photonic and optoelectronic applications remain unexplored. The main obstacle lies in that it is still a challenging issue to produce freestanding borophene with high yield and high purity out of a variety of different phases. In this study, we demonstrate the successful applications of borophene in THz EMW shielding by developing a facile approach for creating composite film consisted of PDMS filled with single-phase and high-crystallinity few-layer $\beta_{12}$-Br nanosheets of high conductivity. Such composite film, which can be scaled up to 5 inches and with tailorable thickness, exhibits an ultrahigh SE and $EES_t$ over 70 dB and $4.8 \times 10^5$ dB·cm$^2$/g, respectively, with only 0.13 wt.% filling ratio of the $\beta_{12}$-Br nanosheets. This successfully circumvents the trade-off between the amount of filling material used and the shielding effectiveness, which usually present in the previous reported EMI shielding materials designed for the THz spectral regime.

Moreover, the remarkable mechanical property of single crystalline $\beta_{12}$-Br nanosheets guarantees the composite film not only handily conforms to different object surfaces with various three-dimensional curvatures, but also produces efficient shielding on these objects. One possible application for our flexible composite film is to prevent the EMW pollution in electrical and optoelectrical circuits from the surrounding environment or adjacent neighborhoods in future 6 G communication networks or large-scale integrated circuits. Besides, with its low-weight, high-efficiency, good-flexibility, and high-stability, and broadband shielding performances, the composite film based on $\beta_{12}$-Br nanosheets could be successfully utilized in a wearable device for daily consumer electronics in the future.

## Methods

### Materials

Boron powders (99.8%, 325 mesh) were purchased from ZhongNuo (China) Co. Ltd. Polydimethylsiloxane (PDMS, SYLGARD 184) and

N-methyl pyrrolidone (NMP, 99.8%) was bought from Dow−Corning (USA) Co. Ltd. and Innochem (China) Co. Ltd, respectively.

### Sample Fabrication

$\beta_{12}$-Br nanosheets are synthesized using our previously developed LTEP method[33]. Firstly, 100-mg boron powders were added into 100-mL NMP solvent, and subsequently stirred for several minutes to form a uniform and well-dispersed solution. Secondly, the mixed solution was transferred into a tip-type ultrasonic system equipped with a cooling system (SXSONIC Incorp., China). The ultrasonic power was maintained at 850 W for exfoliation, which lasted for 10 to 12 h at −20 to −25 °C. Thirdly, the product solution was statically settled at room temperature for 48 to 72 h to thoroughly separate the undissolved boron powders. Finally, the suspension was centrifuged at 11000 revolutions per minute (rpm) for 10 minutes to obtain solid products. And PDMS gel was prepared by mixing the base component with the hardener in a weight fraction of 10:1. The procedures to synthesize $\beta_{12}$-Br/PDMS composite film are as follows. Firstly, the as-grown $\beta_{12}$-Br nanosheet powders were dissolved into 1 mL deionized water via 30 min of bath ultrasonic dispersion, and subsequently added into the PDMS solvent to form the uniform sol by a 1 h of continuous stirring. Secondly, the sol was coated on the surface of laboratory dish and sat for 2 days in air to realize the solidification of the gel. Finally, they were handed into the vacuum chamber and treated at 60 °C for about 12 h to remove the residual moisture. Through the above procedures, the fabrication of large-area $\beta_{12}$-Br/PDMS composite film was accomplished, as shown in Fig. 1b.

### Characterization

The thickness of $\beta_{12}$-Br nanosheets was measured by an atom force microscope (AFM, Bruker Dimension Icon). The morphology and crystalline structure of the nanosheets were investigated by a scanning electron microscope (SEM, Zeis Supra 60) and transmission electron microscope (TEM, FEI Titan 80-300). The scanning transmission electron microscope (STEM), energy disperse X-ray (EDX) mapping and electron energy loss spectroscopy (EELS) techniques were carried out in a JEM ARM200F thermal-field emission microscope with a Cs corrector probe working at 300 kV. For the high-angle annular dark field (HAADF) measurement, a convergence angle of about 21 mrad and collection angle range of 65-172 mrad were adopted for the incoherent atomic number imaging. The chemical compositions were analyzed by XRD patterns recorded on a D-MAX 2200 VPC system. The Raman spectrum of the $\beta_{12}$-Br nanosheets was obtained by inVia Reflex (532 nm laser) made by Renishaw. And the current−voltage characteristics of the borophene nanosheet were tested in an ultra-high vacuum (UHV) probe made by Wavetest.

### Calculation of the crystalline structure of $\beta_{12}$-Br nanosheet

The crystalline structure model of few-layer $\beta_{12}$-Br nanosheet was obtained by the density functional theory (Vienna ab initio simulation package VASP 5.4). Both lattice parameters and atomic positions were optimized using conjugate gradient method, and the convergence criteria for energy and force were $10^{-6}$ eV and $10^{-3}$ eV·Å$^{-1}$, respectively. The kinetic energy cutoff for plane waves was set to 450 eV. The crystal structures were visualized by VESTA package. More details can be found in our previous report[33].

### THz Shielding Measurements

For characterizing the EMI shielding performance of the $\beta_{12}$-Br/PDMS composite thin film, a THz-TDS system is used in our experiments (BATOP TDS 1008). The system operates at room temperature under a $N_2$ atmosphere to minimize absorption by water vapor and other environmental factors. The THz source in this system is a photo-conductive antenna (PCA), which generates broadband THz pulses. Specifically, the THz source is driven by a femtosecond laser with a

pulse duration of <100 fs and a central wavelength at 780 nm. This ultrafast laser pulse is split into two beams: one for THz generation and the other for THz detection. The generation laser pulse is directed onto a PCA, which consists of a low-temperature-grown gallium arsenide patterned with metallic antenna. When the femtosecond laser pulse illuminates the semiconductor, it creates electron−hole pairs, which are then accelerated by an applied bias voltage across the electrodes. This rapid acceleration of charges emits coherent THz pulses with a broadband spectrum, typically ranging from 0.2 to 2 THz.

Broadband THz spectroscopy up to 10 THz was performed using two-color laser-induced air plasma THz generation system combined with air biased coherent detection method. A femtosecond amplifier (Spitfire Ace, Spectra-Physics) was used as the laser source with a pulse width of 35 fs, a center wavelength of 800 nm and a repetition rate of 1 kHz. The samples were attached onto a hollow iron plate for test, and THz wave focused on the sample with a spot radius of 0.5 mm.

The EMI SE of the material can be described using Eq. (1), whereby EMI EES$_t$ of the material can be calculated based on the following equation[15]:

$$\text{EMI EES}_t(\text{dB} \cdot \text{cm}^2 \cdot \text{g}^{-1}) = \frac{(\text{EMI SE}) \cdot S}{m} \qquad (2)$$

where $S$ and $m$ respectively represent the area of the focal spot of THz wave and the mass of $\beta_{12}$-Br nanosheets.

## Statistics and reproducibility

The data in this study were obtained through essential transformation and normalization processes, with outliers excluded based on multiple experimental validations. The data were expressed as mean ± standard deviation, with the sample size indicated in three respective experiments. Statistical analysis was carried out using Origin 2024b.

## THz Nano-imaging

The near-field THz nanoimaging was performed using a commercial near-field THz-TDS (Neaspec, THz-NeaSNOM). To excite the nanosheet, a broadband THz pulse was focused onto an AFM tip (25PtIr200B-H, Rocky Mountain Nanotechnology, 20 nm apex-radius) embedded in scattering-type scanning near-field optical microscope (Supplementary Fig. 13). Specifically, the THz source is driven by a femtosecond laser with a pulse duration in the range of ~100 fs and a central wavelength around 780 nm. The laser pulse is directed onto a PCA patterned onto a low-temperature-grown gallium arsenide patterned with metallic electrodes. When the femtosecond laser pulse illuminates the semiconductor, it creates electron−hole pairs, which are then accelerated by an applied bias voltage across the PCA. This rapid acceleration of charges emits coherent THz pulses with a broadband spectrum, typically ranging from 0.5 to 2 THz. During a specific measurement, the AFM was operated at tapping mode, where the tip vibrated vertically with a frequency of $f = 60$ kHz. The back-scattered light from the tip was collected by a photoconductive antenna receiver. The near-field signal, corresponding to the electric field amplitude $|E_z|$ perpendicular to the substrate, was extracted by demodulating the higher harmonics $nf$ (where $n \geq 3$) of the tip vibration frequency. A typical near-field amplitude image is obtained by integrating $|E_z|$ over the time domain, as shown in Fig. 4i.

## Numerical Simulations

The $\beta_{12}$-Br nanosheet belongs to a metallic 2D material. We approximate the borophene as a 2D conducting layer, with its isotropic conductivity $\sigma$ written as[43,44]:

$$\sigma = \frac{iD}{\pi\left(\omega + \frac{i}{\tau}\right)} \qquad (3)$$

$$D = \frac{\pi e^2 n}{m^*} \qquad (4)$$

where $n$, $\omega$, $\tau$, $D$, and $m^*$ stand for electron charge, electron density, frequency of THz wave, carrier lifetime, Drude weight, and effective electron mass. Once these parameters are determined, using MATLAB R2021b software, the conductivity spectra in the frequency range of 0.1 – 2 THz of $\beta_{12}$-Br nanosheets can be calculated according to Eqs. (3) and (4). With the knowledge of $\sigma$, we employed FDTD (FDTD, Lumerical Solutions Inc.) method to calculate the near-field distribution, charge distribution, and Joule heat loss within an individual borophene circular nanosheet. The THz absorption cross-sections of various individual nanosheets and absorption spectra of the corresponding nanosheet arrays were calculated using the finite element method (COMSOL, MULTIPHYSICS). To simplify the calculation process and conserve computational resources, a perfect 2D disk model without a thickness parameter is used in our calculations.

The numerical simulations on near-field image shown in Fig. 4h was carried out using FDTD analysis. The scanning tip of the scattering-type scanning near-field optical microscope was modeled as a point dipole source raster-scanned across the simulation domain. The dipole, oriented perpendicular to the sample surface, was positioned at a fixed height of 200 nm above the substrate. For each simulation, the perpendicular component of the electric field was extracted in the frequency domain at a plane 100 nm above the surface and converted back to time-domain near-field optical maps via inverse Fourier transform. The permittivity of $n$-doped silicon was adopted from the data in ref. 45, while the dielectric functions of borophene were derived from experimentally measured values.

More details on the numerical simulations and parameters used are given in Supplementary Note 2.

All the three-dimensional models drawn were plotted using the Shapr3D software. All the figures are created using the Adobe Illustrator CC 2017.

## Reporting summary

Further information on research design is available in the Nature Portfolio Reporting Summary linked to this article.

## Data availability

The authors declare that the main data supporting the findings of this study are available within the paper. Source data are provided with this paper (ref. 46). Source data are provided with this paper.

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

## Acknowledgements

H.C. acknowledges support from the National Key Basic Research Program of China (grant nos. 2024YFA1208500 and 2024YFA1208501). F.L. acknowledges support from the National Key Basic Research Program of China (grant no. 2024YFA1207800). S.D. and H.C. acknowledge support from the National Natural Science Foundation of China (grant no. 92463308).

## Author contributions

H.C., S.D., F.L. and N.X. proposed and supervised the projects. H.L. synthesized the materials, characterized their surface morphology and chemical compositions, and carried out the THz shielding performances measurements. H.Z. and Z.C. simulated the absorption spectra of borophene in THz band. X.W. was responsible for the THz imaging of the $\beta_{12}$-Br/PDMS composite film. J.W. created a flowchart illustrating the experimental process. R.Z. provided technical guidance for TEM and XPS testing. H.L., X.W. and H.Z. contributed equally to the work. All the authors involved in the analysis and discussion of the experimental results. And all authors approve to submit the final version of the manuscript.

## Competing interests

The authors declare no competing interests.
