## [Transparent Peer Review file · Nature Communications]

Harmonizing Material Quantity and Terahertz Wave Interference Shielding Efficiency with Metallic Borophene Nanosheets

Corresponding Author: Professor Huanjun Chen

Version 1:

Reviewer comments:

Reviewer #1

(Remarks to the Author)

The manuscript entitled 'Harmonizing Material Quantity and Terahertz Wave Interference Shielding Efficiency with Metallic Borophene Nanosheets' by Haojian Lin et.al. establish a scalable synthesis method for creating a wafer-scale composite of polydimethylsiloxane (PDMS) embedded with few-layer β 12-Br single crystalline nanosheets as conductive fillers. The free electrons of β 12-Br fillers can be excited by THz illumination, dissipating through scattering at the boundaries of the β 12-Br nanosheets. This phenomenon significantly contributes to the exceptional performance of the composite film in THz EMI shielding. The film achieves a mean SE as high as over 70 dB and an ultrahigh EEST of 4.8×10^5 dB-cm³

/g, the highest value to date as we know, across the 0.5 to 2 THz frequency range, with potential extension up to 10 THz. Recently, as a two-dimensional material known as a perfect metal, borophene has attracted wide attention. The concept of the manuscript is novel and will be of interest for the photonics community. However, as given below, some points should be addressed before publication in Nature Communications.

1. The authors claims the trade off between the amount of material used and the absolute EMI shielding effectiveness (EEST) for the EMI shielding materials could be addressed by harnessing the unique properties of two-dimensional (2D) β 12-borophene (β 12-Br) nanosheets. In fact, by progressively increasing the borophene content from 0.13 to 2.1 wt.%, the EMI SE initially experiences rapid augmentation, eventually reaching a plateau for mass ratios over 0.5 wt.%. From this point of view, the previous narrative and the subsequent experimental results are contrary to each other.
2. In order to facilitate the reader to reproduce the contents of the manuscript, the author should provide all simulation parameters, such as the size of the disk, the surrounding refractive index and so on. Since it is necessary to compare with the experiment, is the material parameters set in the simulation process consistent with the experiment? How to ensure consistency? It is particularly noteworthy that for a single borophene disk, how are the transmittance and reflectance, as well as the absorptivity, defined? Generally, the scattering cross section, extinction cross section, and absorption cross section are calculated.
3. The THz absorption spectra of square β 12-Br nanosheets with various thicknesses are presented in Supplementary Fig. 10b, in which the thickness of β 12-Br nanosheets is found to have a weak impact on their THz absorption behaviors. However, in this figure, the author focuses on the thickness of borophene from 1 layer to 700 nm. I think the author should consider further increasing the thickness of borophene to observe whether the absorption spectrum changes. After all, the thickness of the actual preparation is 0.2 to 4 mm, much larger than 700 nm.
4. In Figs 4 f-i, do the author want to compare the simulation results with the experimental results? So, are the resonance wavelengths corresponding to these

pictures consistent ?

Reviewer #2

(Remarks to the Author)

The overall paper is well written and the THz shielding performance of beta12-borophene (Br) nanosheet-based composite sample is explored in terms of shielding efficiency, flexibility, and stability.

The author successfully fabricated a wafer-scale composite of PDMS emvedded with beta12-Br single crystalline nanosheet,

and demonstrated that the proposed material has high flexibility and high effcieincy shielding performance in the THz freuqnecy regime.

While this can be a novel emerging material and nice experimental results, I do not yet see why this work should be published in Nature communications.

The manuscript promises that these approach might once be suitable for suggesting a new EM shielding material.

At the current state of the manuscript, I wouls suggest submitting it to a more Material Sicence and Application journal.

In its current state I cannot recommend acceptance in Nature communications.

Reviewer #3

(Remarks to the Author)

1, this material can achieve 70 dB terahertz EMI shielding efficiency (SE) and 4.8×10^5 dB-cm²/g (@0.87 terahertz) EEST. This is an interesting result. Please explain why this performance is enhanced.

2. The introduction of the manuscript fails to explain the necessity of researching this material and the innovation of the manuscript. The author needs to reconsider the content of the introduction, including the innovation points, the application value of the material, and the outstanding problems existing in the field.

3. It is suggested that the author add a table to compare the work that has been reported.

4. What kind of terahertz wave source does the author use? Please give a detailed description of the experiment and the equipment used.

5. Please explain the basic principle of Figure 4 (f-h).

Version 2:

Reviewer comments:

Reviewer #1

(Remarks to the Author)

Thanks to the author 's reply and the revision of the manuscript, I have no additional questions. I think this version of the manuscript can be accepted now.

Reviewer #3

(Remarks to the Author)

The revision results of the manuscript are satisfactory, and it is recommended to accept.

RESPONSE TO REVIEWERS' COMMENTS

Response to Reviewer #1

We thank this reviewer for the valuable comments on our manuscript.

Comment: The manuscript entitled 'Harmonizing Material Quantity and Terahertz Wave Interference Shielding Efficiency with Metallic Borophene Nanosheets' by Haojian Lin et.al. establish a scalable synthesis method for creating a wafer-scale composite of polydimethylsiloxane (PDMS) embedded with few-layer β_{12} -Br single crystalline nanosheets as conductive fillers. The free electrons of β_{12} -Br fillers can be excited by THz illumination, dissipating through scattering at the boundaries of the β_{12} -Br nanosheets. This phenomenon significantly contributes to the exceptional performance of the composite film in THz EMI shielding. The film achieves a mean SE as high as over 70 dB and an ultrahigh EES_t of 4.8×10^5 dB·cm³/g, the highest value to date as we know, across the 0.5 to 2 THz frequency range, with potential extension up to 10 THz.

Recently, as a two-dimensional material known as a perfect metal, borophene has attracted wide attention. The concept of the manuscript is novel and will be of interest for the photonics community. However, as given below, some points should be addressed before publication in Nature Communications.

Q1: The authors claim the trade-off between the amount of material used and the absolute EMI shielding effectiveness (EES_t) for the EMI shielding materials could be addressed by harnessing the unique properties of two-dimensional (2D) β_{12} -borophene (β_{12} -Br) nanosheets. In fact, by progressively increasing the borophene content from 0.13 to 2.1 wt.%, the EMI SE initially experiences rapid augmentation, eventually reaching a plateau for mass ratios over 0.5 wt.%. From this point of view, the previous narrative and the subsequent experimental results are contrary to each other.

R1: We thank the reviewer for pointing out this interesting yet important issue. Our experimental results shown in Figure 4b of the original manuscript are not contradictory to our claim that "the trade-off between the amount of material used and the absolute EMI shielding effectiveness (EES_t) for the EMI shielding materials could be addressed."

As pointed out by the reviewer, by progressively increasing the borophene content from 0.13 to 2.1 wt.%, the EMI SE initially experiences rapid augmentation, eventually reaching a plateau for mass ratios over 0.5 wt.%. Such saturation effect can be attributed to the competition between absorption and scattering of incident THz wave by the β_{12} -Br nanosheets at different borophene content. According to our previous study (Front. Mater. 2021, 8, 737347.) and that from other research group (Nat. Mater. 2013, 12, 304.), THz wave absorption by the β_{12} -Br nanosheets will monotonically increase with the borophene content. When the borophene content arrives at a critical value, the maximum THz absorption of the β_{12} -Br nanosheets will be achieved. But if the borophene content is continuously increased to exceed the critical value, the scattering coefficient

will sharply increase, originating from the strong scattering loss of the incident THz waves by the nanosheet assemblies inside the PDMS matrix instead of being fully absorbed by β_{12} -Br nanosheets. This will lead to suppression of THz wave absorption and therefore saturation of the EMI SE.

This behavior can be quantitatively described using **coupled mode theory** (Nat. Mater. 2013, 12, 304.; Front. Mater. 2021, 8, 737347.), which describes the absorption, reflection, and transmission spectra of the β_{12} -Br nanosheets as a function of their contents within the PDMS matrix. Figure R1 shows a schematic diagram of the borophene nanosheets dispersed in a PDMS matrix. To simplify the discussion, the nanosheets are approximated as rectangular array. Here, $\varepsilon_1 = \varepsilon_2$ represent the dielectric functions of the PDMS separated by the interface.

Figure R1. Schematic diagram showing the calculation of the absorption, reflection, and transmission spectra of the β_{12} -Br nanosheets embedded into the PDMS matrix using the coupled mode theory.

The perpendicularly incident THz waves can be expressed as $|s_+\rangle = [s_{1+} s_{2+}]^T$, while the outgoing THz waves can be written as $|s_-\rangle = [s_{1-} s_{2-}]^T$, where s_{1+} (s_{1-}) and s_{2+} (s_{2-}) are the complex amplitudes of the incident (outgoing) wave in the PDMS matrix. The time-varying mode amplitude, a , of the β_{12} -Br sheets can be written as,

$$\begin{cases} \frac{da}{dt} = j(\omega_0 + j\Gamma_{tot}/2)a + \langle d |^* \rangle s_+ \\ |s_-\rangle = C |s_+\rangle + a |d\rangle \end{cases} \quad (R1)$$

where the total decay rate (Γ_{tot}) is the sum of THz radiative decay rate (Γ_{rad}) and the THz absorption rate (Γ_{abs}), and $|d\rangle = [d_1 d_2]^T$ is the coupling vector under the condition of $\langle d | d \rangle = \Gamma_{rad}$. Parameter C is the background scattering of PDMS matrix without the β_{12} -Br nanosheets, which

can be depicted as,

$$C = \begin{bmatrix} r_0 & jt_0 \\ jt_0 & r_0 \end{bmatrix} \quad (\text{R2})$$

where $r_0 = \frac{\sqrt{\varepsilon_2} - \sqrt{\varepsilon_1}}{\sqrt{\varepsilon_1} + \sqrt{\varepsilon_2}}$ and $t_0 = \frac{2\sqrt{\varepsilon_1\varepsilon_2}}{\sqrt{\varepsilon_1} + \sqrt{\varepsilon_2}}$ are the background reflection and transmission coefficients, respectively. According to the time-reversal symmetry and conservation of energy, $CC^+ = \mathbf{I}$ and $C|d\rangle^* = -|d\rangle$, parameters d_1 and d_2 can be expressed as,

$$\begin{cases} d_1 = -\sqrt{\frac{\Gamma_{rad}(1-r_0)}{2}} \\ d_2 = j\sqrt{\frac{\Gamma_{rad}(1+r_0)}{2}} \end{cases} \quad (\text{R3})$$

By solving the above differential equation, the time-varying mode amplitude (a) can be written as,

$$a = \frac{\langle |d|^* \rangle s_+}{j(\omega - \omega_0 - j\Gamma_{tot}/2)} \quad (\text{R4})$$

When the THz wave incidents vertically onto the interface between PDMS and β_{12} -Br nanosheets from the upper space of the PDMS matrix, the reflection (r) and transmission (t) coefficients of β_{12} -Br nanosheets can be obtained as,

$$\begin{cases} r = \frac{s_{1-}}{s_{1+}} = r_0 + \frac{\Gamma_{rad}(1-r_0)/2}{j(\omega - \omega_0) + \Gamma_{tot}/2} \\ t = \frac{s_{2-}}{s_{1+}} = jt_0 - \frac{j\Gamma_{rad}t_0/2}{j(\omega - \omega_0) + \Gamma_{tot}/2} \end{cases} \quad (\text{R5})$$

Therefore, the total absorption, A , of the β_{12} -Br nanosheets embedded into the PDMS matrix can be deduced as (Front. Mater. 2021, 8, 737347.),

$$A = 1 - |r|^2 - |t|^2 = \frac{|a|^2 \Gamma_{abs}}{|s_{1+}|^2} = \frac{\Gamma_{abs} \Gamma_{rad} (1-r_0)/2}{(\omega - \omega_0)^2 + (\Gamma_{tot}/2)^2} \quad (\text{R6})$$

According to Equation R6, the absorption coefficient A of the β_{12} -Br nanosheets is a function of

the radiative decay rate and absorption decay rate. For nanosheets with sizes much smaller than the wavelength of the THz wave, which is the case in our current study, the radiative decay rate can be approximated as (Front. Mater. 2021, 8, 737347.),

$$\Gamma_{rad} = \frac{\zeta_1^2 E_F e^2}{2\pi\epsilon_0 c \bar{n} \hbar^2} \frac{D^2}{S} \quad (R7)$$

where e is the charge of an electron, \hbar is the reduced Planck constant, \bar{n} is the average refractive index of the surrounding medium, ζ_1 is approximated as a constant for the longitudinal plasmonic dipole mode, E_F is Fermi energy, c is the speed of light, ϵ_0 is the vacuum permittivity, $D = \sqrt{LW}$ is the characteristic size of the borophene nanosheet, **S is the lattice area with $S = \mathbf{P}_x \mathbf{P}_y$** . Therefore, Γ_{rad} is determined by the duty ratio, D^2/S , of β_{12} -Br nanosheet arrays, which is proportional to the borophene weight fraction. The maximum absorption is achieved under the condition $\partial A / \partial \Gamma_{rad} = 0$, yielding the well-known critical coupling condition $\Gamma_{rad} = \Gamma_{abs}$. Consequently, when measuring the absorption of the β_{12} -Br/PDMS composite, it initially increases and reaches a maximum value once the borophene content corresponding to the critical coupling condition is achieved. Beyond this point, the absorption strength, which is proportional to the EMI SE, begins to decrease. This qualitatively explains the experimental results shown in Figure 4b of the original manuscript, where the THz EMI SE of the β_{12} -Br/PDMS film initially increases with borophene content and then gradually saturates.

Another reason for the saturation behavior can be attributed to the detection limit of the terahertz time-domain spectroscopy (THz-TDS) system used in our study. According to the shielding

efficiency equation, EMI SE (dB) = $20 \log_{10} \left(\frac{I_{in}}{I_{out}} \right)$, where I_{in} and I_{out} represent the intensities of

the incident and transmitted THz waves, respectively. When the borophene content reaches approximately 0.5 wt.%, the THz EMI SE achieves a value of up to 80 dB. This corresponds to **99.99%** of the incident wave being absorbed, with only 0.01% of the wave transmitting through the composite film. The detection limit of the THz-TDS spectrometer used in our measurements (BATOP, TDS 1008) is 85 dB. This means that when the transmitted wave intensity falls below $5 \times 10^{-3}\%$ of the incident wave, the spectrometer can no longer reliably distinguish intensity differences associated with varying transmittance levels. As a result, the THz EMI SE of the β_{12} -Br/PDMS film appears to saturate in our experiments once the borophene content exceeds 0.5 wt.%, remaining nearly constant with further increases in borophene content.

A discussion on clarifying the saturation of EMI SE against the borophene weight fraction is added in paragraph 1 in page 11 as,

"...Specifically, by progressively increasing the borophene content from 0.13 to 2.1 wt.%, the EMI SE initially experiences rapid augmentation, eventually reaching a plateau for weight fraction over

0.5 wt.%. Such saturation effect can be attributed to the competition between absorption and scattering of incident THz wave by the β_{12} -Br nanosheets at different borophene content. According to previous studies^{38, 39}, THz wave absorption by the β_{12} -Br nanosheets will first monotonically increase with the borophene content. When the borophene content arrives at a critical value, the maximum THz absorption of the β_{12} -Br nanosheets will be achieved. But if the borophene content is continuously increased to exceed the critical value, the scattering coefficient will sharply increase, originating from the strong scattering loss of the incident THz waves by the nanosheet assemblies inside the PDMS matrix instead of being fully absorbed by β_{12} -Br nanosheets. This will lead to suppression of THz wave absorption and therefore saturation of the EMI SE (see more detail discussion in Supplementary Note 1 and Supplementary Fig. 8).

Another reason for the saturation behavior can be attributed to the detection limit of the THz-TDS system used in our study. According to the shielding efficiency equation, EMI SE (dB) = $20 \log_{10} \left(\frac{I_{in}}{I_{out}} \right)$, where I_{in} and I_{out} represent the intensities of the incident and transmitted THz waves, respectively. When the borophene content reaches approximately 0.5 wt.%, the THz EMI SE achieves a value of up to 80 dB. This corresponds to 99.99% of the incident wave being absorbed, with only 0.01% of the wave transmitting through the composite film. The detection limit of the THz-TDS spectrometer used in our measurements (BATOP, TDS 1008) is 85 dB. This means that when the transmitted wave intensity falls below $5 \times 10^{-3}\%$ of the incident wave, the spectrometer can no longer reliably distinguish intensity differences associated with varying transmission levels. As a result, the THz EMI SE of the β_{12} -Br/PDMS film appears to saturate in our experiments once the borophene content exceeds 0.5 wt.%, remaining nearly constant with further increases in borophene content."

The detail discussion on critical coupling of the β_{12} -Br/PDMS film, i.e., the aforementioned discussion on Figure R1 and Equation (R1) to (R7), is added as Note S1 in the Supplementary Information part. Figure R1 is added as Figure S8 in the Supplementary Information part.

Q2: *In order to facilitate the reader to reproduce the contents of the manuscript, the author should provide all simulation parameters, such as the size of the disk, the surrounding refractive index and so on. Since it is necessary to compare with the experiment, is the material parameters set in the simulation process consistent with the experiment? How to ensure consistency? It is particularly noteworthy that for a single borophene disk, how are the transmittance and reflectance, as well as the absorptivity, defined? Generally, the scattering cross section, extinction cross section, and absorption cross section are calculated.*

R2: We appreciate the reviewer's suggestion on the numerical simulations in our study. We employed the finite-difference time-domain (FDTD, Lumerical) method to calculate the near-field distribution, charge distribution, and Joule heat loss within an individual borophene circular

nanosheet, as shown in Figure 4g–4h in our original manuscript. According to the experimental measurement shown in Figure 4i in our original manuscript, the diameter of the nanosheet is set as 5.64 μm . To simplify the calculation process and conserve computational resources, a perfect 2D disk model without a thickness parameter is used in our calculations. The nanosheet is placed onto 500- μm -thick highly doped n -Si substrate, exactly as that used in the experimental near-field measurement. The surrounding environment is set as vacuum, with a refractive index of 1.0.

One pivotal parameter is the complex conductivity of borophene. However, to our best knowledge, there is no literature reports on the conductivity of borophene in the THz domain. Considering that our synthesized β_{12} -Br nanosheets exhibit a very high conductance, we employed Drude model, which describes the electromagnetic responses of free electrons, to calculate the conductivity of borophene, which is assumed to be isotropic and written as (Opt. Express 2020, 28, 16725.),

$$\sigma = \frac{iD}{\pi\left(\omega + \frac{i}{\tau}\right)}, D = \frac{\pi e^2 n}{m^*} \quad (\text{R8})$$

where n , ω , τ , D , and m^* stand for electron charge, electron density, frequency of THz wave, carrier lifetime, Drude weight, and effective electron mass.

As an initial approximation, m^* is adopted as $m^* = 1.4 \times m_0 = 1.274 \times 10^{-30}$ kg. The carrier density n and relaxation time τ are adopted from the theoretical studies, with values of $3.4 \times 10^{19} \text{ m}^{-2}$ (J. Am. Chem. Soc. 2017, 139, 17181.) and 65 fs (Opt. Express 2020, 28, 16725.), respectively. The resulting conductivity is depicted in Figure R2a and R2b (black curves). The subsequent step involves comparing the simulation results with experimental data to assess their consistency. To this end, we calculate the THz absorption spectra of a disk array using the finite element method (COMSOL MULTIPHYSICS). Based on morphological characterization (Figure 1c in the original manuscript), most borophene nanosheets exhibit irregular shapes. Therefore, a square disk array is employed as a simplified model for the simulation, as illustrated in Figure R2c. Specifically, the side length of each disk is set to 5 μm , a value determined by the measured average lateral size of the nanosheets, as illustrated in Figure S3b of the Supplementary Information section of our original manuscript. This choice ensures that the model accurately reflects the physical dimensions of the synthesized β_{12} -Br nanosheets. The separation between adjacent disks is set to 20 μm to avoid potential near-field electromagnetic coupling. The disks are arranged in a 2D square lattice. The surrounding refractive index is set as 1.53 to model the PDMS matrix. Using the initial conductivity, the calculated absorption spectrum of the square disk array reveals a resonance around 8.7 THz (Figure R2d), which significantly deviates from the experimental measurements. The experimental data for the β_{12} -Br/PDMS composite film show strong THz absorption around 0.87 THz (Figure 4a in the original manuscript). This discrepancy suggests that the conductivity parameters need to be adjusted to achieve better agreement between the simulation and experimental results.

Figure R2. (a) and (b) Real (a) and imaginary (b) part of β_{12} -Br conductivity at different correction coefficient ζ , respectively. (c) Schematic showing the simulation model. (d) Simulated THz absorption spectrum of 2D β_{12} -Br nanosheet array, where τ is 65 fs, $n = 3.4 \times 10^{19} \text{ m}^{-2}$. (e) THz absorption spectra of 2D β_{12} -Br nanosheet arrays using the conductivities at different ζ . The arrays are all placed inside a PDMS environment.

Given that the β_{12} -Br nanosheets in our study were synthesized using a wet chemical method, unintentional doping and structural defects are likely to be introduced. These factors will result in carrier density and relaxation time values that differ from those previously reported. Our goal is to redshift the resonance frequency of the nanosheet array to a lower range, which requires a reduction in electron density, as supported by plasmonic theory (*Plasmonics: Fundamentals and Applications*, Stefan A. Maier, 2007, Springer New York, NY.). To achieve this, we introduce a correction factor ζ to adjust the electron density, expressed as $n = 3.4 \times 10^{19}/\zeta \text{ cm}^{-2}$. Reducing n will consequently lower the electron scattering rate, leading to a corresponding adjustment in the relaxation time as $\tau = 65\zeta$. This approach allows us to fine-tune the conductivities (Figure R2a and R2b, color lines) to better align the simulated resonance frequency with the experimental observations. **As shown in Figure R2d and R2e, the resonance peak of the β_{12} -Br nanosheet arrays shifts from 8.7 THz to approximately 0.88 THz at $\zeta = 180$, which is consistent with the experimental result (0.87 THz).** Therefore, a ζ of 180 is employed in all of the simulations in our manuscript.

Once the conductivity has been determined, we can proceed to calculate the THz wave responses of nanosheets with varying shapes and sizes. This includes evaluating the absorption cross-section, near-field distribution, charge distribution, and Joule loss within an individual nanosheet. Figure R3 presents the calculated absorption cross-sections for nanosheets with different shapes (Figure

R3a) and sizes (Figure R3b). The results demonstrate that the β_{12} -Br nanosheets exhibit strong resonances originated from the free electrons, whereby the resonance frequency is highly sensitive to its size and shape, allowing for tunability across a broad spectral range. In addition, for nanosheet arrays, the maximum absorption efficiency remains nearly constant at 43%, regardless of the geometrical parameters of the borophene nanosheets (Figure R3c). Consequently, the broadband yet relatively stable THz EMI absorption shielding performance observed in our experimental measurements for the β_{12} -Br/PDMS composite film can be attributed to the presence of nanosheets with varying thicknesses and shapes within the composite film. This diversity in nanosheet morphology contributes to the consistent and wide-ranging absorption characteristics of the material.

Figure R3. (a) Calculated absorption cross-sections for nanosheets with circular, hexagon, rectangle, and square shapes. The areas of these nanosheets are set as $25 \mu\text{m}^2$. (b) Calculated absorption cross-sections for square nanosheets of different side lengths. (c) Calculated absorption spectra of nanosheet arrays with different shapes. The areas of the nanosheets are all set as $25 \mu\text{m}^2$. The separations between adjacent nanosheets in these arrays are set as $20 \mu\text{m}$.

The THz near-field characteristics of a specific nanosheet can also be readily calculated once its absorption cross-section spectrum is obtained. According to the experimental measurement shown in Figure 4i (Figure R4d) in the main text, the diameter of the disk is set as $5.64 \mu\text{m}$. The disk is placed onto $500\text{-}\mu\text{m}$ -thick highly doped n -Si substrate, exactly as that used in the experimental near-field measurement. The surrounding environment is set as vacuum, with a refractive index of 1.0. The refractive index (\tilde{n}) of n -doped silicon was written as (Meas. Sci. Technol. 2022, 13, 1727.),

$$\tilde{n}^2 = \varepsilon_{\text{Si}} - \frac{\omega_p^2}{\omega(\omega + i\Gamma)}, \quad \omega_p^2 = \frac{N_c e^2}{\varepsilon_0 m_{\text{Si}}^*} \quad (\text{R9})$$

where ε_{Si} , Γ , and N_c equal 11.66, 6.141 THz, and $2 \times 10^{16} \text{ cm}^{-3}$, respectively. Parameter ε_0 is the vacuum permittivity, m_{Si}^* is adopted as $m_{\text{Si}}^* = 0.26 \times m_0 = 2.368 \times 10^{-31} \text{ kg}$. To calculate the near-field distribution and compare it with the experimental near-field measurement, the scanning tip of the scattering-type scanning near-field optical microscope was modeled as a point dipole

source raster-scanned across the simulation domain. The dipole, oriented perpendicular to the sample surface, was positioned at a fixed height of 200 nm above the substrate. For each simulation, the perpendicular component of the electric field, $|E_z|$, was extracted in the frequency domain (0.5 to 2.0 THz) at a plane 100 nm above the surface and converted back to time-domain near-field optical maps via inverse Fourier transform. The $|E_z|$ was then integrated over the time domain to obtain the near-field distribution, as shown in Figure 4h (i.e., Figure R4c below) in the main text of our original manuscript. For calculation of the charge density (ρ) and Joule loss (J) within the nanosheet, a broadband plane wave was employed to illuminate the nanosheet, whereby the electric field, \vec{E} , within the nanosheet was recorded. Afterwards, ρ and J can be obtained according to,

$$\rho = \epsilon_0 \nabla \cdot \vec{E} \quad (\text{R10})$$

$$J = \frac{1}{2} \sigma \vec{E} \cdot \vec{E}^* \quad (\text{R11})$$

To ensure consistency with the experimental conditions, we employed the same broadband source (0.5 to 2.0 THz) in our simulations to excite an individual borophene nanosheet and performed spectral integration of the ρ and J over the same spectral range.

For a circular nanosheet, the THz illumination induces strong electron oscillations within the nanosheet, which are subsequently reflected by the disk boundary (Figure R4a). This results in significant Joule loss occurring within the borophene nanosheets (Figure R4b), leading to attenuated electromagnetic fields inside the disk compared to the substrate (Figure R4c). This phenomenon is further corroborated by THz near-field optical measurements conducted on an individual β_{12} -Br nanosheet (Figure R4d). As a result, the Joule loss contributes to a remarkably high absorption efficiency of up to 43% for the nanosheet array at resonance, even for a monolayer thickness (Figure R3c). This highlights the efficient energy dissipation mechanism enabled by the rich free electrons within the β_{12} -Br nanosheets.

Figure R4. (a, b, c) Simulated results of the spatial distributions of localized electric field (a), electron charge density (b), and Joule loss (c) for an individual β_{12} -Br nanosheet in frequency ranging from 0.5 to 2 THz, respectively. (d) Experimental THz near-field image of a single β_{12} -Br nanosheet, captured in frequency range of 0.5 to 2 THz. Both the simulation and experimental measurement utilize a 500- μm -thick highly doped n -silicon substrate. The diameters of the nanosheets are all set at 5.64 μm .

According to the suggestion by the reviewer, we added the details on the numerical simulations as Note S2 in the Supplementary Information part. Figure R2 and R3 are added as Figure S9 and S10 in the Supplementary Information part. A table (Table R1) listing the main parameters used in the simulations is also added as Table S2 in the Supplementary Information part.

Table R1. Parameters used in the numerical simulations.

Parameters	Values
n	$1.89 \times 10^{17} \text{ m}^{-2}$
τ	11.7 ps
m^*	$1.274 \times 10^{-30} \text{ kg}$
Refractive index of PDMS	1.5331
Refractive index of the surrounding vacuum	1
Permittivity of the highly doped n -Si substrate (ϵ_{Si})	11.66
Side length of square disk used in the absorption spectra simulations shown in Figure R2d	5.0 μm
Side length of square disk used in the absorption spectra simulations shown in Figure R2e	5.0 μm
Diameter of circular disk used in the near-field responses simulations shown in Figure R4a, R4b, and R4c	5.64 μm
Diameter of circular disk used in the absorption cross-sections simulations shown in Figure R3a	5.64 μm

Side length of square disk used in the absorption cross-sections simulations shown in Figure R3a	5.0 μm
Side length of hexagon disk used in the absorption cross-sections simulations shown in Figure R3a	3.10 μm
Side length of rectangle disk used in the absorption cross-sections simulations shown in Figure R3a	10 μm \times 2.5 μm
Side length of square disk used in the absorption cross-sections simulations shown in Figure R3b	3 to 7 μm , with an increment of 1 μm
Diameter of circular disk used in the absorption spectra simulations shown in Figure R3c	5.64 μm
Side length of square disk used in the absorption spectra simulations shown in Figure R3c	5.0 μm
Side length of hexagon disk used in the absorption spectra simulations shown in Figure R3c	3.10 μm
Side length of rectangle disk used in the absorption spectra simulations shown in Figure R3c	10 μm \times 2.5 μm
Separation between adjacent disks in the array	20 μm
Thicknesses of all of the disks used in simulations of Figure R2 to R4	0

In response to the reviewer's comments regarding the THz absorption of individual borophene nanosheets, we have calculated the absorption cross-sections of individual nanosheets with various shapes and sizes to illustrate their THz resonances, which arise from the free electrons. These results are presented in Figure R3a and R3b. Additionally, to demonstrate the macroscopic THz absorption properties of the β_{12} -Br/PDMS composite films, we have included the absorption spectra of nanosheet arrays, as shown in Figure R2d and R2e.

In the revised manuscript, the discussion on the numerical simulation of THz absorption of borophene disks and their arrays in paragraph 2 in page 13 is modified as,

"This mechanism can be further supported by simulating the dynamics of electrons and the corresponding absorption of EMW energy within a β_{12} -Br nanosheet (Methods and Supplementary Note 2). The behavior of free electrons in β_{12} -Br nanosheet in response to the EMW is governed by the alternating-current conductivity, which can be characterized using the Drude model (Supplementary Fig. 9). For a nanosheet with a monolayer thickness, electromagnetic resonance characterized by strong absorption peaks can be excited upon irradiation by the THz wave (Supplementary Fig. 9). These resonances induce strong electron oscillations within the nanosheet, which are subsequently reflected by the nanosheet boundary (Fig. 4f). This results in significant Joule loss occurring within the borophene nanosheets (Fig. 4g), leading to attenuated electromagnetic fields inside the nanosheet compared to the substrate (Fig. 4h). This phenomenon is further corroborated by THz near-field optical measurements conducted on an individual β_{12} -Br nanosheet (Fig. 4i, see Method for numerical simulations). As a result, the Joule loss contributes to a remarkably high absorption efficiency of up to 43% for the nanosheet array at resonance, even for a monolayer thickness (Supplementary Fig. 9). This highlights the efficient energy dissipation mechanism enabled by the rich free electrons within the β_{12} -Br nanosheets.

It should be noted that the experimental THz nano-imaging was conducted on the samples using a scattering-type THz optical microscope (THz-NeaSNOM, Neaspec GmbH, see Method for details). The nano-imaging system is equipped with a broadband THz operating in pulse mode, which covers a frequency range of 0.5 to 2.0 THz. Due to the limitations of this light source, we were only able to conduct broadband near-field imaging. As a result, Fig. 4i represents the outcome obtained through broadband acquisition of the near-field THz wave amplitude. In the simulations, to ensure consistency with the experimental conditions, we employed the same broadband source (0.5 to 2.0 THz) in our simulations to excite an individual borophene nanosheet and performed spectral integration of the electron concentration (Fig. 4f) and Joule loss (Fig. 4g) over the same spectral range. For the near-field THz wave amplitude, in the simulation the perpendicular component of the electric field, $|E_z|$, was extracted in the frequency domain (0.5 to 2.0 THz) at a plane 100 nm above the surface and converted back to time-domain near-field optical maps via inverse Fourier transform. The $|E_z|$ was then integrated over the time domain to obtain the near-field distribution, as shown in Fig. 4h (see Supplementary Note 2 for more simulation details).

According to the simulation results, the resonance frequency of the β_{12} -Br nanosheet is strongly dependent on its size, shape, and thickness, covering a broad spectral range (Supplementary Fig. 10a and Fig. 11a). The maximum absorption efficiency remains nearly constant at 43% (Supplementary Fig. 10c), regardless of the shapes of the borophene nanosheets, while the lateral size of the nanosheet strongly influences the absorption cross-section of the individual nanosheet (Supplementary Fig. 10b). The THz absorption intensity is also influenced by the nanosheet thickness. Specifically, when the borophene thickness is ≤ 500 nm (Supplementary Fig. 11b), the absorption of the array stabilizes at 44%, which is comparable to that of a monolayer borophene nanosheet array (43%). As the nanosheet thickness increases further, a reduction in absorption intensity is observed. Given that the thicknesses of the borophene nanosheets used in our study is less than 5 nm (Fig. 1c and Supplementary Fig. 3a), the broadband yet relatively stable THz EMI absorption shielding performance observed in our experimental measurements for the β_{12} -Br/PDMS composite film can be attributed to the presence of nanosheets with varying sizes and shapes within the composite film."

Figure R4a, R4b, and R4c are added as Figure 4f, 4g, and 4h, respectively.

In the Method part of the revised manuscript, the description on numerical simulations is also modified.

Q3: The THz absorption spectra of square β_{12} -Br nanosheets with various thicknesses are presented in Supplementary Fig. 10b, in which the thickness of β_{12} -Br nanosheets is found to have a weak impact on their THz absorption behaviors. However, in this figure, the author focuses on the thickness of borophene from 1 layer to 700 nm. I think the author should consider further

increasing the thickness of borophene to observe whether the absorption spectrum changes. After all, the thickness of the actual preparation is 0.2 to 4 mm, much larger than 700 nm.

R3: In our original manuscript, Supplementary Figure 10b is intended to demonstrate the influence of the thickness of the β_{12} -Br nanosheet on the THz absorption of the nanosheet arrays. In contrast, Figure 4d in the main text is intended to discuss the influence of the thickness of the β_{12} -Br/PDMS composite thin film on its THz EMI absorption shielding performance. Therefore, these two figures refer to different sample thicknesses: Supplementary Figure 10b pertains to the thickness of a single β_{12} -Br nanosheet (monolayer to 700 nm), while Figure 4d in the main text pertains to the thickness of the macroscopic composite thin film (0.2 to 4 mm). We apologize for any misunderstanding that may have arisen.

As suggested by the reviewer, we calculate the absorption spectra of β_{12} -Br nanosheet arrays with different nanosheet thicknesses to demonstrate the absorption spectrum changes against the nanosheet thickness. To that end, we calculate the three-dimensional (3D) dielectric function of the borophene nanosheet according to (Opt. Mater. Express 2021, 11, 2627.),

$$\varepsilon_r = \varepsilon_\infty - \frac{e^2 n}{m^* \varepsilon_0 d \left(\omega^2 + \frac{1}{\tau^2} \right)}, \quad \varepsilon_i = \frac{e^2 n / \tau}{m^* \varepsilon_0 d \omega \left(\tau \omega^2 + \frac{1}{\tau^2} \right)} \quad (\text{R12})$$

where ε_r and ε_i are respectively the real and imaginary parts of the complex dielectric function, $\varepsilon_\infty = 11$ is the relative permittivity, $\varepsilon_0 = 8.854 \times 10^{-12} \text{ F} \cdot \text{m}^{-1}$ is the vacuum permittivity, and d represents the thickness of β_{12} -Br nanosheet. Equation R9 gives the thickness-dependent dielectric function of the β_{12} -Br nanosheet, whereby the THz absorption spectra of the nanosheet arrays with different borophene thicknesses can be readily calculated.

Figure R5. (a) Simulated results of the absorption spectra of β_{12} -Br nanosheet arrays of different nanosheet thicknesses. (b) Enlarged spectra in the frequency range of 0.7 to 0.9 THz. The nanosheets are of square shape and arranged into square lattice, with side length of 5 μm . The separation between adjacent nanosheets are 20 μm . The arrays are placed inside

PDMS matrix.

As shown in Figure R5, the THz absorption intensity and peak are dependent on the nanosheet thicknesses. Specifically, as the thickness is increased, the absorption resonance frequency redshifts towards the low frequency region (Figure R5a). In addition, when the borophene thickness is ≤ 500 nm (Figure R5b), the absorption of the array stabilizes at 44%, comparable to that of a monolayer borophene nanosheet array (43%). When the nanosheet thickness is further increased, a reduction of absorption intensity is observed. Considering that the thicknesses of the borophene nanosheets used in our study is less than 5 nm (Figure 1c and Figure S3a of the original manuscript), we claim that the thickness of β_{12} -Br nanosheets used in our study is found to have a weak impact on their THz absorption behaviors.

As suggested by the reviewer, a discussion regarding the influence of nanosheet thickness on the THz absorption behavior of the nanosheet array is added in paragraph 2 in page 14 as,

"According to the simulation results, the resonance frequency of the β_{12} -Br nanosheet is strongly dependent on its size, shape, and thickness, covering a broad spectral range (Supplementary Fig. 10a and Fig. 11a). The maximum absorption efficiency remains nearly constant at 43% (Supplementary Fig. 10c), regardless of the shapes of the borophene nanosheets, while the lateral size of the nanosheet strongly influences the absorption cross-section of the individual nanosheet (Supplementary Fig. 10b). The THz absorption intensity is also influenced by the nanosheet thickness. Specifically, when the borophene thickness is ≤ 500 nm (Supplementary Fig. 11b), the absorption of the array stabilizes at 44%, which is comparable to that of a monolayer borophene nanosheet array (43%). As the nanosheet thickness increases further, a reduction in absorption intensity is observed. Given that the thicknesses of the borophene nanosheets used in our study is less than 5 nm (Fig. 1c and Supplementary Fig. 3a), the broadband yet relatively stable THz EMI absorption shielding performance observed in our experimental measurements for the β_{12} -Br/PDMS composite film can be attributed to the presence of nanosheets with varying sizes and shapes within the composite film."

In addition, the discussion on the dependence of THz-wave shielding performance of the β_{12} -Br/PDMS composite film on its thickness (paragraph 2 in page 12) is modified as,

"The THz-wave shielding performance of the composite film is also highly dependent on its thickness, as evidenced by the EMI SE spectra plotted against the composite film thickness (Fig. 4c). If the weight fraction of borophene nanosheets was kept at 0.5 wt.%, both of the EMI SE and EES_t values monotonically increase with the composite film thickness (Fig. 4d). Particularly, for a composite film thickness of 4 mm, the maximum EMI SE and EES_t can reach as high as 85 dB and 2.5×10^5 dB·cm²·g⁻¹ at 0.87 THz, respectively. These results evidently demonstrate the exceptional EMW shielding performance of the β_{12} -Br/PDMS composite film."

Figure R5 is added as Figure S11 in the Supplementary Information part.

Q4: In Figs 4 f-i, do the author want to compare the simulation results with the experimental results? So, are the resonance wavelengths corresponding to these pictures consistent?

R4: As the reviewer noted, in Figure 4f to 4i of our original manuscript, we aimed to compare the simulation results with the experimental data. In our experimental setup, the nano-imaging system is equipped with a broadband THz time-domain spectroscopy (THz-TDS) source, which operates within a frequency range of 0.5 to 2.0 THz. Due to the limitations of this light source, we were only able to conduct broadband near-field imaging. As a result, Figure 4i (now referred to as Figure R6i below) in the original manuscript represents the outcome obtained through broadband acquisition. Specifically, Figure 4i corresponds to the electric field amplitude $|E_z|$ perpendicular to the substrate, which was extracted by demodulating the higher harmonics nf (where $n \geq 3$) of the tip vibration frequency, and then obtained by integrating $|E_z|$ over the time domain. In the simulations, to ensure consistency with the experimental conditions, we employed the same broadband source (0.5 to 2.0 THz) in our simulations to excite an individual borophene disk and performed spectral integration of the near-field THz wave intensity, yielding the result shown in Figure R6h. Furthermore, we integrated the electron concentration and Joule loss over the same spectral range to generate the results presented in Figure R6f and R6g.

In the revised manuscript, Figure 4 has been replaced with Figure R6, as illustrated below.

Figure R6. EMI shielding properties of β_{12} -Br/PDMS composite film. (a) The EMI SE curves of the composite film with various weight fractions of β_{12} -Br nanosheets. (b) The curves of EMI SE/EMI EES_t to the weight fraction of borophene nanosheets at 0.87 THz. (c) The frequency-dependent EMI SE curves of the composite film with different thickness. (d) The relationship between EMI SE/EMI EES_t and film thickness at 0.87 THz. (e) Schematic showing the excitation of electron oscillations in β_{12} -Br nanosheets under THz radiation. (f and g) Simulated spatial distributions of charge density (f) and Joule loss (g) within an individual β_{12} -Br nanosheet. (h and i) Simulated (h) and experimental (i) near-field image of an individual β_{12} -Br nanosheet on a 500- μ m-thick highly doped *n*-silicon substrate. The simulations and experimental measurements are conducted in the frequency range of 0.5 – 2 THz. The diameters of the circular nanosheets are all set at 5.64 μ m, as determined by the experimental measurement shown in (i). (j) The EMI SE stability of the β_{12} -Br/PDMS composite film after 15-day air storage.

A discussion on this issue is added in paragraph 3 in page 13 as,

"It should be noted that the experimental THz nano-imaging was conducted on the samples using a scattering-type THz optical microscope (THz-NeaSNOM, Neaspec GmbH, see Method for details). The nano-imaging system is equipped with a broadband THz source operating in pulse mode, which covers a frequency range of 0.5 to 2.0 THz. Due to the limitations of this light source, we were only able to conduct broadband near-field imaging. As a result, Fig. 4i represents the outcome obtained through broadband acquisition of the near-field THz wave amplitude. In the simulations, to ensure consistency with the experimental conditions, we employed the same broadband source (0.5 to 2.0 THz) in our simulations to excite an individual borophene nanosheet and performed spectral integration of the electron concentration (Fig. 4f) and Joule loss (Fig. 4g) over the same spectral range. For the near-field THz wave amplitude, in the simulation the perpendicular component of the electric field, $|E_z|$, was extracted in the frequency domain (0.5 to 2.0 THz) at a plane 100 nm above the surface and converted back to time-domain near-field optical maps *via* inverse Fourier transform. The $|E_z|$ was then integrated over the time domain to obtain the near-field distribution, as shown in Fig. 4h (see Supplementary Note 2 for more simulation details)."

Response to Reviewer #2

We thank this reviewer for the rigorous comments on our manuscript and insightful suggestions.

Comment: The overall paper is well written and the THz shielding performance of beta12-borophene (Br) nanosheet-based composite sample is explored in terms of shielding efficiency, flexibility, and stability.

The author successfully fabricated a wafer-scale composite of PDMS embedded with beta12-Br single crystalline nanosheet, and demonstrated that the proposed material has high flexibility and high efficiency shielding performance in the THz frequency regime.

While this can be a novel emerging material and nice experimental results, I do not yet see why this work should be published in Nature communications.

The manuscript promises that these approach might once be suitable for suggesting a new EM shielding material.

At the current state of the manuscript, I would suggest submitting it to a more Material Science and Application journal.

In its current state I cannot recommend acceptance in Nature communications.

R: We sincerely thank the reviewer for her/his thoughtful evaluation of our manuscript and for acknowledging the novelty and experimental results of our work. We are, however, sorry to hear that the reviewer feels our study does not meet the criteria for publication in *Nature Communications*. While we understand and respect the reviewer's assessment, we would like to respectfully argue that our work represents a significant advancement in the field of functional materials for THz wave interference shielding.

Our research leverages the unique properties of single-crystalline β_{12} -Br nanosheets, specifically their superhigh-density free electrons and high conductivity, to address a critical and long-standing challenge in EMI shielding materials: the trade-off between material usage and achieving high absolute EMI shielding effectiveness (EES_t) and shielding efficiency (SE). Traditional materials often require large thicknesses or high filler loadings to achieve sufficient shielding performance, which compromises flexibility, stability, and practical applicability. In contrast, our work demonstrates that the proposed β_{12} -Br/PDMS composite achieves exceptional THz shielding performance with minimal material usage, while maintaining high flexibility and stability.

Importantly, this study represents the **first report on the THz shielding application of β_{12} -Br nanosheets**, a novel and highly promising material. We emphasize that our work marks the pioneering synthesis of a 5-inch β_{12} -Br/PDMS composite film with exceptional THz shielding capabilities, surpassing all existing shielding materials reported in recent research (please see Table R2 below). **Specifically, we simultaneously achieve a THz EMI SE of 70 dB and an EES_t of 4.8×10^5 dB·cm²/g using the β_{12} -Br/PDMS composite. This surpasses the values of previously reported THz shielding materials with an EES_t less than 3×10^5 dB·cm²/g and a SE smaller than 60 dB, while only needs 0.1 wt.% of these materials to realize the same SE value.** As such, our study advances the borophene material system—previously at the forefront of cutting-edge science but underexplored in applied research (e.g., Synthesis of borophane polymorphs through hydrogenation of borophene, *Science* 2021, 371, 1143.; Synthesis of borophenes: Anisotropic, two-dimensional boron polymorphs, *Science* 2015, 350, 1513.; Micrometre-scale single-crystalline borophene on a square-lattice Cu(100) surface, *Nature Chemistry* 2022, 14, 377.)—toward practical utilization. By demonstrating its exceptional performance in THz shielding, we not only highlight the material's unique properties but also bridge the gap between fundamental material science and practical applications. This work thus provides a clear pathway for the material science community to further explore and utilize β_{12} -Br in advanced technologies, particularly in the rapidly growing fields of THz communication, imaging, and sensing.

Table R2. Comparison of EMI SE performances of various THz shielding materials.

Materials	Weight ratios (wt.%)	Thickness (mm)	Frequency (THz)	EMI SE (dB)	EMI EES _t (dB·cm ³ ·g ⁻¹)	Ref.
MXene film	–	0.0005	0.25–2.00	30	–	Adv. Opt. Mater. 2018, 6, 1701076
MWCNTs film	–	~0.02	0.40–2.20	30	–	J. Opt. Soc. Am. B 2016, 33, 2430-2436
α -Br/PDMS	~33.33	5	0.10–2.70	~42	0.125	ACS Appl. Mater. Interfaces 2020, 12, 19746-19754
MXene/PAA/ACC	8.5	0.13	0.20–2.00	45.3	–	ACS Nano 2021, 15, 1465-1474
α -Br tablet	–	1	0.10–2.70	50	–	J. Mater. Sci. Technol. 2020, 52, 136-144
Gr/Cu	–	0.00016	0.10–1.00	60.95	–	Nanotechnology 2020, 31, 505710
MXene film	–	0.025	0.30–0.70	70	–	J. Mater. Sci.: Mater. Electron. 2018, 29, 17245-17253
rGO paper	–	~0.37	0.10–1.00	72.1	–	Appl. Phys. Lett. 2008, 93, 231905
OLC/PMMA	2	0.12	0.1–3	4	3.4	Opt. Lett. 2014, 39, 1541-1544
SWCNT film on PET film	–	5×10 ⁻⁸	0.1–1.2	4	3.6	Diam. Relat. Mater. 2012, 25, 13-18
SWCNTs/PVA	1.6	0.3	0.3–2.1	29	15.7	Appl. Phys. Lett. 2011, 98, 174101
MWCNTs/PMMA	~2	~0.48	0.10–4.00	20	16.9	Nanoscale 2018, 10, 13426-13431
CNF/PTFE/PVDF/PMMA	~10	~0.05	0.57–0.63	32	27.1	Appl. Phys. Lett. 2011, 98, 174101
Gr/PDMS	10	~2	0.10–0.80	~32	30	Nanoscale 2018, 10, 13426-13431
CNW/PMC	50	~0.03	0.57–0.63	~40	36.4	Appl. Phys. Lett. 2012, 101, 910
PAN/PU	10	0.15	0.2–1.2	42	38.2	Electron. Lett. 2007, 43, 1271-1273
Graphite/PMMA	35.7	0.30	0.20–0.90	~50	42.4	J. Appl. Phys. 2006, 99, 066103
Kapton-derived carbon	–	0.125	0.22–0.5	~70	54.5	Carbon 2016,

						100, 158-164
Zn ²⁺ /MXene/GO foams	–	0.085	0.20–2.00	~51	451	ACS Nano 2020, 14, 2109-2117
GO/Fe ₃ O ₄	500	10	0.10–2.50	–	16000	ACS Appl. Mater. Interfaces 2019, 11, 1274-1282
Gr-1500/MWCNT	–	3	0.10–1.60	40	21000	Adv. Opt. Mater. 2018, 6, 1801165
MXene/GO foams	20	4	0.20–2.00	–	46000	Adv. Opt. Mater. 2018, 6, 1801165
Gr foams-1500	–	3	0.10–1.60	50	110000	ACS Appl. Mater. Interfaces 2019, 11, 25369-25377
Gr/PMMA	–	0.034	0.1–1.00	60	300000	Nat. Commun. 2021, 12, 4655
3D Gr	–	–	0.1–3	35	30000	Adv. Electron. Mater., 2024, 10, 2300853
Gr/TiCN @ Polyimide/MXene	–	0.016	0.5–1.5	45	900	ACS Appl. Nano Mater. 2023, 6, 23401-23409
Fe ₃ O ₄ /Gr	27.3	10	0.1–2.5	42	16000	ACS Appl. Nano Mater. 2023, 6, 5264-5273
Fe ₃ O ₄ /Polymer/CNFs	40	0.54	0.1–1.2	~60	–	Adv. Sci. 2024, 11, 2305099
MXene/CNT Janus	–	0.00005	0.3–1.6	60	230756	ACS Appl. Mater. Interfaces 2022, 14, 57008-57015
MXene/PGPDMS	83.3	0.12	0.5–3	57.5	24000	Adv. Funct. Mater. 2024, 34, 2400732
Aramid/MXene Janus	80	0.9	0.2–2.4	60.49	–	Adv. Sci. 2024, 11, 2305898
Polymer/Cellulose	50	6.9	0.2–1.2	~58	66493.99	Adv. Mater. Interfaces 2023, 10, 2300440
3D MXene/PI	80	0.11	0.6–1.1	70.4	–	J. Appl. Polym. Sci. 2023, 140, e53790
CNFs/Polyme	50	~0.6	0.1–1.2	43.9	–	Adv. Funct. Mater. 2023, 33, 2210578

Gr/Fe ₃ GeTe ₂ /FeTe ₂ /Fe ₃ Ge	–	2.5	0.5–1.6	76	–	Small Methods 2023, 7, 2201493
Gr-220/700	5	2	0.2–1.2	56.6	–	J. Appl. Polym. Sci. 2022, 139, e52511
PA/TS@IL-Ag-rGO	3	–	0.2–2	36	–	Adv. Optical Mater. 2022, 10, 2101868
PUS-Ni/MXene	~53	8	0.15–2.15	69.8	65.8	J. Mater. Chem.2022, A 10, 23570-23579
MXene/PPy	91	0.04	0.2–1.6	71.4	36983	Mater. Chem. Phys. 2024, 311, 128573.
MXene/Gr	75	0.048	0.1–1	60	–	44
CNF/SBC	20	3	0.4–2	70	–	Chem. Eng. J. 2023, 467, 143213
OCF/GO	100	4.34	0.3–1.5	34	–	Carbon 2022, 199, 333-346
MXene-PMMA /rGO- PVP	16.7	0.148	0.37–2	57.7	–	Carbon 2022, 194, 127-139
MS-Ni/CNT	51.6	8	0.1–2.2	79.1	66.3	Chem. Eng. J. 2022, 429, 132393
β_{12}-Br/PDMS	0.13 0.50	2 4	0.10–2.00	68 83	480000 250000	This work

Note: MXene (Ti₃C₂T_x), GO (graphene oxide), PAA (poly (acrylic acid)), ACC 189 (amorphous calcium carbonate), CNF (carbon nano fiber), PTFE 190 (polytetrafluoroethylene), PVDF (poly (vinylidene fluoride)), PMMA (poly (methyl 191 methacrylate)), CNW (carbon nano whiskers), PMC (fluor acrylic copolymer), EBA 192 (ethylene co-butyl acrylate), MWCNTs (multiwalled carbon nanotubes), PVA 193 (polyvinyl alcohol), PDMS (polydimethylsiloxane), OLC (onion-like carbon), PAN 194 (polyaniline), PU (polyurethane), SBC (sustainable biocarbon), OCF (oxidized carbon 195 fiber), MS (melamine sponge).

Here is more detail explanation on the novelties and important achievements of our study:

THz EMI shielding materials are in great demand for future applications in information communication, healthcare, and mineral resource exploration. However, current THz EMI shielding materials face a trade-off between low material usage and achieving high absolute EMI EES_t and SE. This trade-off has limited previously reported THz shielding materials (such as metallic films, MXene composites, graphene, etc., please see Table R2 above) to exhibit an EES_t below 3×10^4 dB·cm²/g and an SE less than 60 dB. Additionally, the increasing demand of wearable devices and portable electronics in consumer electronics underscores the need for EMI materials that are cost-effective, easily manufacturable, lightweight, and highly stretchable to conform to various surfaces.

In the current study, we develop a composite film of PDMS embedded with few-layer β_{12} -Br single-crystalline nanosheets as conductive fillers to address the aforementioned trade-off and requirements. Here are the key points highlighting the novelty and significance of our manuscript.

1) Achieving a mean SE exceeding 70 dB and setting a benchmark with an EES_t of 4.8×10^5 dB·cm³/g—the highest reported value to date—across a broad frequency range from 0.5 to 2 THz. Current research in this field is focused on exploring polymer-based EMI shielding materials incorporating segregated conductive fillers with high free electron density and large carrier mobility, such as MXene and graphene. However, the finite carrier densities in these materials restrict their EES_t and SE to below 3×10^5 dB·cm²/g and 60 dB, respectively.

By leveraging the superhigh-density free electrons ($\sim 3.4 \times 10^{19}$ m⁻²) and high conductivity of single-crystalline β_{12} -Br nanosheets, we have employed these nanosheets as conductive fillers in PDMS thin films. **Our composite film achieves a remarkable EES_t of 4.8×10^5 dB·cm³/g and a mean SE exceeding 70 dB in an ultrabroad range from 0.1 to 2 THz, while utilizing only 1/1000 and 1/100 of the mass required by MXene and graphene (e.g., Nanoscale 2018, 10, 13426.; J. Mater. Chem. A 2022, 10, 23570.), respectively.** Our result also demonstrates the potential for extension these excellent shielding performances up to 10 THz.

2) Developing a 5-inch composite film of PDMS embedded with few-layer β_{12} -Br single-crystalline nanosheets as conductive fillers, with thickness ranging from 0.2 to 4 mm. Although few-layer borophene nanosheets exhibit outstanding electrical and thermal properties (npj 2D Mater. Appl. 2017, 1, 14.), their application in THz EMI shielding can't meet the actual shielding requirements due to the absence of the fabricating technique of large-area EMI shielding thin films. For instance, a recent study used α -borophene nanoflakes as conductive fillers in PDMS films, but due to the inefficient EMI shielding of α -Br, an ultrahigh concentration of borophene—**up to 100 wt.%**—was required (ACS Appl. Mater. Interfaces 2020, 12, 19746.). This high concentration precludes achieving uniform and continuous large-area films due to cracking in the PDMS matrix.

In our study, by leveraging the superior EES_t of the β_{12} -Br, we successfully embedded an exceptionally low borophene content of less than 0.5 wt.% in the PDMS matrix while maintaining outstanding shielding performance of the composite film. This significant reduction in β_{12} -Br content facilitates homogeneous dispersion within the PDMS matrix, enabling the formation of flexible, large-area thin films **up to 5 inches**.

3) By capitalizing on the β_{12} -Br/PDMS composite's superior mechanical properties, including 158% tensile strain at a Young's modulus of 33 MPa, we demonstrate the high-efficiency shielding performance of conformably coated surfaces based on β_{12} -Br nanosheets. These findings suggest the great potential of this material in the EMI shielding area.

To sum up, we believe this breakthrough is of broad interest to the scientific community, as it not only introduces a novel material but also provides a scalable and practical solution for advanced THz shielding applications. The ability to fabricate wafer-scale composites with single-crystalline β_{12} -Br nanosheets further underscores the potential for real-world implementation. Given the growing importance of THz technology in communication, imaging, and sensing, we argue that our findings have the potential to significantly impact both fundamental research and applied materials science.

To strengthen the presentation of our study, we have revised the Introduction section of our manuscript to emphasize the outstanding challenges in the field of THz EMI shielding materials, the necessity of researching β_{12} -Br, the key innovations of our work, and the application potential of β_{12} -Br.

In paragraph 3 in page 3, the discussion on the outstanding problems in the field of THz EMI shielding materials is modified as,

"Despite significant advances in the field, a critical issue remains: the trade-off between material mass and shielding performance. Specifically, current THz shielding materials exhibit EEST values below 3×10^4 dB·cm²/g and SE values under 60 dB (Supplementary Table 1)¹⁵. Improving performance often necessitates increasing the filler content or material thickness, which in turn compromises mechanical stability and restricts applicability in lightweight, flexible devices. Moreover, for practical deployment, there is an urgent need for environmentally sustainable and scalable manufacturing processes for these materials—yet this remains a significant challenge^{2, 21}. These persistent difficulties highlight the pressing demand for new materials that can combine high EEST and SE values with flexibility, low density, and scalable production."

In paragraph 4 in page 3, the discussion on the necessity of researching β_{12} -Br is modified as,

"Borophene, the boron analogue of graphene, emerges as a promising candidate to meet these demands. As the lightest metalloid²², boron imparts borophene with ultralow density, rendering it inherently lightweight—a critical advantage for applications in portable and wearable electronics. Moreover, borophene exhibits a high carrier density ($3.3 \times 10^{19} \sim 4.3 \times 10^{19}$ m⁻²)²³ and exceptional electron mobility ($878.6 \sim 28.4 \times 10^5$ cm² V⁻¹ s⁻¹)^{24, 25}, properties essential for achieving strong THz absorption while minimizing material usage. These characteristics enable borophene to effectively dissipate EMW through scattering and re-absorption by free electrons, significantly reducing the quantity of material required to achieve high shielding performance. In addition to its electronic properties, structural versatility of borophene^{22, 26}—manifested in multiple phases such as triangular, honeycomb, stripe, and rectangular configurations—endows it with exceptional flexibility and mechanical strength, as evidenced by its high Young's modulus (170 ~ 398 GPa·nm)²⁶. This flexibility makes borophene an ideal filler for polymer matrices, enabling it to conform to arbitrary surfaces and meet the demands of advanced EMI shielding applications in flexible electronics. By leveraging the unique combination of ultralow density, high carrier density,

and outstanding electron mobility in borophene, it is possible to overcome the trade-off between material quantity and THz EMI performance. This positions borophene as a transformative material for next-generation EMI shielding, offering a pathway to achieve high SE with minimal material usage while retaining the flexibility required for practical applications. However, despite these advantages, the development of borophene-based EMI shielding materials has been severely hindered by the lack of a scalable synthesis method for high-yield, freestanding metallic borophene. As a result, reported EES_t values remain below $0.125 \text{ dB} \cdot \text{cm}^2/\text{g}$, and SE values are lower than 42 dB ^{27, 28}—far from realizing its full potential. "

In paragraph 2 in page 4, the discussion on the innovation points of our study and the application value of β_{12} -Br is modified as,

"In this study, we address the above limitations by developing a scalable synthesis method for wafer-scale polydimethylsiloxane (PDMS) composites embedded with few-layer β_{12} -Br single-crystalline nanosheets, which exhibit extraordinary free electron density and excellent electrical conductivity ($3.0 \times 10^4 \text{ S/m}$). Due to strong THz resonances initiated by the free electrons, the composite achieves a mean SE of over 70 dB and an unprecedented EES_t of $4.8 \times 10^5 \text{ dB} \cdot \text{cm}^3/\text{g}$ —the highest reported value to date—across the 0.5 to 2 THz range, with potential extension up to 10 THz. These results surpass those of previously reported THz EMI shielding materials, while utilizing only 1/10000th to 1/100th of the mass required by other materials (Supplementary Table 1). Furthermore, the composite exhibits remarkable flexibility, with a tensile strain exceeding 158% at a tensile stress of 33 MPa, enabling effective shielding on conformably coated surfaces. Based on the β_{12} -Br/PDMS composites, we further demonstrate their practical applications in EMW shielding of objects with irregular surfaces. This work not only demonstrates the potential of few-layer metallic β_{12} -Br nanosheets as highly efficient, low-density, and elastic materials for THz shielding but also addresses critical challenges in the field. By providing a scalable synthesis method and achieving record-breaking performance, we pave the way for borophene-based composites in next-generation EMI shielding applications, particularly in portable, wearable, and miniaturized devices where lightweight, flexibility, and high performance are paramount."

The benchmarking of β_{12} -Br in terms of THz EMI shielding and mechanical properties against other reported materials is provided in Supplementary Table 1 (Table R2 above) and Supplementary Table 3 (Table R3 below) in the Supplementary Information part.

Response to Reviewer #3

We thank this reviewer for the valuable comments and suggestions on our manuscript.

Q1: *This material can achieve 70 dB terahertz EMI shielding efficiency (SE) and $4.8 \times 10^5 \text{ dB} \cdot \text{cm}^2/\text{g}$ (@0.87 terahertz) EES_t . This is an interesting result. Please explain why this*

performance is enhanced.

R1: We sincerely thank the reviewer for this insightful question regarding the enhanced THz EMI shielding performance of our β_{12} -Br nanosheet-based composite. The exceptional SE of 70 dB and the EES_t of 4.8×10^5 dB \cdot cm 2 /g (@0.87 THz) can be attributed to the electronic and structural properties of β_{12} -Br nanosheets. Each single-crystalline β_{12} -Br nanosheet supports a remarkably high concentration of free electrons (3.4×10^{19} /m 2) with exceptionally large mobility (2.84×10^6 cm 2 /(V \cdot s)) (J. Am. Chem. Soc. 2017, 139, 17181.), which are among the highest reported values for 2D materials. When exposed to THz waves, these free electrons are excited and accelerated by the electric field of the EMW, resulting in collective electron oscillations within the nanosheets. The planar size of the β_{12} -Br nanosheets (approximately 5 μ m, as shown in Figure 1c and Figure S3b of our original manuscript) is significantly smaller than the wavelength of the incident THz radiation, causing the excited electrons to encounter the nanosheet boundaries and undergo multiple reflections. These reflections act as a restoring force, giving rise to strong resonances in the THz spectral region, as demonstrated in Figure R3 and Figure R5. Due to the subwavelength size of the nanosheets, most of these resonances decay via absorption rather than scattering (Figure R7), a phenomenon well-explained by the principles outlined in *Absorption and Scattering of Light by Small Particles* (Craig F. Bohren and Donald R. Huffman, 2008, John Wiley & Sons.). As a result, the kinetic energy of the oscillating electrons is efficiently transferred to the lattice of the β_{12} -Br nanosheets, as illustrated in Figure R6g, converting nearly all of the absorbed EMW into heat. This energy dissipation process, driven by the high electron mobility, confinement-induced resonances, and dominant absorption mechanism, leads to the ultrahigh EMI absorption effectiveness observed in our study.

Figure R7. Calculated absorption (purple) and scattering (blue) cross-sections for nanosheets with circular, hexagon, rectangle, and square shapes. The areas of these nanosheets are set as 25 μ m 2 . The scattering cross-sections are multiplied by 100 times for a better comparison with the absorption cross-sections, showing that the scattering of an individual nanosheet is much smaller than its absorption.

A discussion on the mechanism of the superior THz EMI performances of the β_{12} -Br/PDMS

composite thin film in our study is added in paragraph 3 in page 12 as,

"Afterwards, the mechanism governing the strong THz EMW absorption of the β_{12} -Br/PDMS composite film are further explored. The preceding discussion unambiguously suggests that the incident EMW are efficiently absorbed and dissipated by various β_{12} -Br nanosheets within the composite film. This can be attributed to the electronic and structural properties of β_{12} -Br nanosheets. Each single-crystalline β_{12} -Br nanosheet supports a remarkably high concentration of free electrons with exceptionally large mobility²⁵, which are among the highest reported values for 2D materials. When exposed to THz waves, these free electrons are excited and accelerated by the electric field of the EMW, resulting in collective electron oscillations within the nanosheets. The planar size of the β_{12} -Br nanosheets (approximately 5 μm , as shown in Fig. 1c and Supplementary Fig. 3b) is significantly smaller than the wavelength of the incident THz radiation, causing the excited electrons to encounter the nanosheet boundaries and undergo multiple reflections. These reflections act as a restoring force, giving rise to strong resonances in the THz spectral region (Supplementary Fig. 10 and Fig. 11). Due to the subwavelength size of the nanosheets, most of these resonances decay via absorption rather than scattering (Supplementary Fig. 12), a phenomenon well-explained by the principles of electromagnetic absorption and scattering by small particles⁴⁰. As a result, the kinetic energy of the oscillating electrons is efficiently transferred to the lattice of the β_{12} -Br nanosheets, converting nearly all of the absorbed EMW into heat. This energy dissipation process, driven by the high electron mobility, confinement-induced resonances, and dominant absorption mechanism, leads to the ultrahigh EMI absorption effectiveness observed in our study."

The Figure R7 is added as Supplementary Fig. 12 in the Supplementary Information part. A reference "*Absorption and Scattering of Light by Small Particles*, Craig F. Bohren and Donald R. Huffman, 2008, John Wiley & Sons." is added as Ref. 40 in reference list.

Q2: The introduction of the manuscript fails to explain the necessity of researching this material and the innovation of the manuscript. The author needs to reconsider the content of the introduction, including the innovation points, the application value of the material, and the outstanding problems existing in the field.

R2: We sincerely thank the reviewer for her/his valuable suggestion.

Our research leverages the unique properties of single-crystalline β_{12} -Br nanosheets, specifically their superhigh-density free electrons and high conductivity, to address a critical and long-standing challenge in EMI shielding materials: the trade-off between material usage and achieving high absolute EES_t and SE. Traditional materials often require large thicknesses or high filler loadings to achieve sufficient shielding performance, which compromises flexibility, stability, and practical applicability. In contrast, our work demonstrates that the proposed β_{12} -Br/PDMS composite

achieves exceptional THz shielding performance with minimal material usage, while maintaining high flexibility and stability.

Importantly, this study represents the **first report on the THz shielding application of β_{12} -Br nanosheets**, a novel and highly promising material. We emphasize that our work marks the pioneering synthesis of a 5-inch β_{12} -Br/PDMS composite film with exceptional THz shielding capabilities, surpassing all existing shielding materials reported in recent research (please see Table R2 above). **Specifically, we simultaneously achieve a THz EMI SE of 70 dB and an EES_t of 4.8×10^5 dB·cm²/g using the β_{12} -Br/PDMS composite. This surpasses the values of previously reported THz shielding materials with an EES_t less than 3×10^5 dB·cm²/g and a SE smaller than 60 dB, while only needs 0.1 wt.% of these materials to realize the same SE value.** As such, our study advances the borophene material system—previously at the forefront of cutting-edge science but underexplored in applied research (e.g., Synthesis of borophane polymorphs through hydrogenation of borophene, *Science* 2021, 371, 1143.; Synthesis of borophenes: Anisotropic, two-dimensional boron polymorphs, *Science* 2015, 350, 1513.; Micrometre-scale single-crystalline borophene on a square-lattice Cu(100) surface, *Nature Chemistry* 2022, 14, 377.)—toward practical utilization. By demonstrating its exceptional performance in THz shielding, we not only highlight the material's unique properties but also bridge the gap between fundamental material science and practical applications. This work thus provides a clear pathway for the material science community to further explore and utilize β_{12} -Br in advanced technologies, particularly in the rapidly growing fields of THz communication, imaging, and sensing.

As suggested by the reviewer, we modify the Introduction part in our revised manuscript to highlight the outstanding problems existing in the field of THz EMI shielding materials, the necessity of researching β_{12} -Br, the innovation points, and the application value of the β_{12} -Br.

In particular, in paragraph 3 in page 3, the discussion on the outstanding problems in the field of THz EMI shielding materials is modified as,

" Despite significant advances in the field, a critical issue remains: the trade-off between material mass and shielding performance. Specifically, current THz shielding materials exhibit EES_t values below 3×10^4 dB·cm²/g and SE values under 60 dB (Supplementary Table 1)¹⁵. Improving performance often necessitates increasing the filler content or material thickness, which in turn compromises mechanical stability and restricts applicability in lightweight, flexible devices. Moreover, for practical deployment, there is an urgent need for environmentally sustainable and scalable manufacturing processes for these materials—yet this remains a significant challenge^{2, 21}. These persistent difficulties highlight the pressing demand for new materials that can combine high EES_t and SE values with flexibility, low density, and scalable production."

In paragraph 4 in page 3, the discussion on the necessity of researching β_{12} -Br is modified as,

" Borophene, the boron analogue of graphene, emerges as a promising candidate to meet these

demands. As the lightest metalloid²², boron imparts borophene with ultralow density, rendering it inherently lightweight—a critical advantage for applications in portable and wearable electronics. Moreover, borophene exhibits a high carrier density ($3.3 \times 10^{19} \sim 4.3 \times 10^{19} \text{ m}^{-2}$)²³ and exceptional electron mobility ($878.6 \sim 28.4 \times 10^5 \text{ cm}^2 \text{ V}^{-1} \text{ s}^{-1}$)^{24, 25}, properties essential for achieving strong THz absorption while minimizing material usage. These characteristics enable borophene to effectively dissipate EMW through scattering and re-absorption by free electrons, significantly reducing the quantity of material required to achieve high shielding performance. In addition to its electronic properties, structural versatility of borophene^{22, 26}—manifested in multiple phases such as triangular, honeycomb, stripe, and rectangular configurations—endows it with exceptional flexibility and mechanical strength, as evidenced by its high Young's modulus ($170 \sim 398 \text{ GPa} \cdot \text{nm}$)²⁶. This flexibility makes borophene an ideal filler for polymer matrices, enabling it to conform to arbitrary surfaces and meet the demands of advanced EMI shielding applications in flexible electronics. By leveraging the unique combination of ultralow density, high carrier density, and outstanding electron mobility in borophene, it is possible to overcome the trade-off between material quantity and THz EMI performance. This positions borophene as a transformative material for next-generation EMI shielding, offering a pathway to achieve high SE with minimal material usage while retaining the flexibility required for practical applications. However, despite these advantages, the development of borophene-based EMI shielding materials has been severely hindered by the lack of a scalable synthesis method for high-yield, freestanding metallic borophene. As a result, reported EES_t values remain below $0.125 \text{ dB} \cdot \text{cm}^2/\text{g}$, and SE values are lower than 42 dB ^{27, 28}—far from realizing its full potential. "

In paragraph 2 in page 4, the discussion on the innovation points of our study and the application value of $\beta_{12}\text{-Br}$ is modified as,

"In this study, we address the above limitations by developing a scalable synthesis method for wafer-scale polydimethylsiloxane (PDMS) composites embedded with few-layer $\beta_{12}\text{-Br}$ single-crystalline nanosheets, which exhibit extraordinary free electron density and excellent electrical conductivity ($3.0 \times 10^4 \text{ S/m}$). Due to strong THz resonances initiated by the free electrons, the composite achieves a mean SE of over 70 dB and an unprecedented EES_t of $4.8 \times 10^5 \text{ dB} \cdot \text{cm}^2/\text{g}$ —the highest reported value to date—across the 0.5 to 2 THz range, with potential extension up to 10 THz . These results surpass those of previously reported THz EMI shielding materials, while utilizing only $1/10000$ th to $1/100$ th of the mass required by other materials (Supplementary Table 1). Furthermore, the composite exhibits remarkable flexibility, with a tensile strain exceeding 158% at a tensile stress of 33 MPa , enabling effective shielding on conformably coated surfaces. Based on the $\beta_{12}\text{-Br}/\text{PDMS}$ composites, we further demonstrate their practical applications in EMW shielding of objects with irregular surfaces. This work not only demonstrates the potential of few-layer metallic $\beta_{12}\text{-Br}$ nanosheets as highly efficient, low-density, and elastic materials for THz shielding but also addresses critical challenges in the field. By providing a scalable synthesis method and achieving record-breaking performance, we pave the way for borophene-based composites in next-generation EMI shielding applications, particularly in portable, wearable, and miniaturized devices where lightweight, flexibility, and high performance are paramount."

The benchmarking of β_{12} -Br in terms of THz EMI shielding and mechanical properties against other reported materials is provided in Supplementary Table 1 (Table R2 above) and Supplementary Table 3 (Table R3 below) in the Supplementary Information part.

Q3: It is suggested that the author add a table to compare the work that has been reported.

R3: As suggested by the reviewer, we have added two tables to the Supplementary Information section. Table S1 (i.e., Table R2 above) compares the THz EMI shielding performance of our β_{12} -Br/PDMS composite thin film to that of other reported THz EMI shielding materials. Additionally, Table S3 (i.e., Table R3 below) provides a comparison of the mechanical performance of our β_{12} -Br/PDMS composite thin film with other reported THz EMI shielding materials based on 2D materials.

Table R3. Tensile properties of different 2D material-based composites for THz EMI shielding.

Materials	Tensile Stress (Pa)	Strain (%)	Ref.
MXene/PAA/ACC hydrogels	20k	1500	ACS Nano 2021, 15, 1465-1474
MXene/PVA hydrogels	100k	50	ACS Mater. Lett. 2022, 4, 2352-2361
Gr/PMMA	~48M	~8	Nat. Commun. 2021, 12 , 4655
GO/PMMA	~26M	~2.2	J. Mater. Sci. 2013, 48, 6223-6232
β_{12} -Br/PDMS	~32M	~158	This work

Note: MXene ($\text{Ti}_3\text{C}_2\text{T}_x$), GO (graphene oxide), PAA (poly (acrylic acid)), ACC (amorphous calcium carbonate), Gr (graphene).

Q4: What kind of terahertz wave source does the author use? Please give a detailed description of the experiment and the equipment used.

R4: For characterizing the EMI shielding performance of the β_{12} -Br/PDMS composite thin film, a THz-TDS system is used in our experiments (BATOP TDS 1008). The system operates at room temperature under a N_2 atmosphere to minimize absorption by water vapor and other environmental factors. The THz source in this system is a photoconductive antenna (PCA), which generates broadband THz pulses. Specifically, the THz source is driven by a femtosecond laser with a pulse duration of <100 fs and a central wavelength at 780 nm. This ultrafast laser pulse is split into two beams: one for THz generation and the other for THz detection. The generation laser pulse is directed onto a PCA, which consists of a low-temperature-grown gallium arsenide

patterned with metallic antenna. When the femtosecond laser pulse illuminates the semiconductor, it creates electron–hole pairs, which are then accelerated by an applied bias voltage across the electrodes. This rapid acceleration of charges emits coherent THz pulses with a broadband spectrum, typically ranging from 0.2 to 2 THz.

Broadband THz spectroscopy up to 10 THz was performed using two-color laser-induced air plasma THz generation system combined with air biased coherent detection method. A femtosecond amplifier (Spitfire Ace, Spectra-Physics) was used as the laser source with a pulse width of 35 fs, a center wavelength of 800 nm and a repetition rate of 1 kHz. The samples were attached onto a hollow iron plate for test, and THz wave focused on the sample with a spot radius of 0.5 mm.

The near-field THz nanoimaging was performed using a commercial near-field THz-TDS (Neaspec, THz-NeaSNOM). To excite the nanosheet, a broadband THz pulse was focused onto an AFM tip (25PtIr200B-H, Rocky Mountain Nanotechnology, 20 nm apex-radius) embedded in scattering-type scanning near-field optical microscope (Figure R8). Specifically, the THz source is driven by a femtosecond laser with a pulse duration in the range of ~ 100 fs and a central wavelength around 780 nm. The laser pulse is directed onto a PCA patterned onto a low-temperature-grown gallium arsenide patterned with metallic electrodes. When the femtosecond laser pulse illuminates the semiconductor, it creates electron–hole pairs, which are then accelerated by an applied bias voltage across the PCA. This rapid acceleration of charges emits coherent THz pulses with a broadband spectrum, typically ranging from 0.5 to 2 THz. During a specific measurement, the AFM was operated at tapping mode, where the tip vibrated vertically with a frequency of $f = 60$ kHz. The back-scattered light from the tip was collected by a photoconductive antenna receiver. The near-field signal was extracted by demodulating the higher harmonics nf ($n \geq 3$) of the tip vibration frequency. A typical near-field amplitude image is obtained by integrating $|E_z|$ over the time domain.

Figure R8. Schematic showing operation principle of the near-field THz-TDS system.

As suggested by the reviewer, we provide a detailed description of the β_{12} -Br nanosheets and their composites in Method part as,

"THz Shielding Measurements. For characterizing the EMI shielding performance of the β_{12} -Br/PDMS composite thin film, a THz-TDS system is used in our experiments (BATOP TDS 1008). The system operates at room temperature under a N_2 atmosphere to minimize absorption by water vapor and other environmental factors. The THz source in this system is a photoconductive antenna (PCA), which generates broadband THz pulses. Specifically, the THz source is driven by a femtosecond laser with a pulse duration of <100 fs and a central wavelength at 780 nm. This ultrafast laser pulse is split into two beams: one for THz generation and the other for THz detection. The generation laser pulse is directed onto a PCA, which consists of a low-temperature-grown gallium arsenide patterned with metallic antenna. When the femtosecond laser pulse illuminates the semiconductor, it creates electron-hole pairs, which are then accelerated by an applied bias voltage across the electrodes. This rapid acceleration of charges emits coherent THz pulses with a broadband spectrum, typically ranging from 0.2 to 2 THz.

Broadband THz spectroscopy up to 10 THz was performed using two-color laser-induced air plasma THz generation system combined with air biased coherent detection method. A femtosecond amplifier (Spitfire Ace, Spectra-Physics) was used as the laser source with a pulse width of 35 fs, a center wavelength of 800 nm and a repetition rate of 1 kHz. The samples were attached onto a hollow iron plate for test, and THz wave focused on the sample with a spot radius of 0.5 mm.

THz Nano-imaging. The near-field THz nanoimaging was performed using a commercial near-field THz-TDS (Neaspec, THz-NeaSNOM). To excite the nanosheet, a broadband THz pulse was focused onto an AFM tip (25PtIr200B-H, Rocky Mountain Nanotechnology, 20 nm apex-radius) embedded in scattering-type scanning near-field optical microscope (Supplementary Fig. 13). Specifically, the THz source is driven by a femtosecond laser with a pulse duration in the range of ~ 100 fs and a central wavelength around 780 nm. The laser pulse is directed onto a PCA patterned onto a low-temperature-grown gallium arsenide patterned with metallic electrodes. When the femtosecond laser pulse illuminates the semiconductor, it creates electron-hole pairs, which are then accelerated by an applied bias voltage across the PCA. This rapid acceleration of charges emits coherent THz pulses with a broadband spectrum, typically ranging from 0.5 to 2 THz. During a specific measurement, the AFM was operated at tapping mode, where the tip vibrated vertically with a frequency of $f = 60$ kHz. The back-scattered light from the tip was collected by a photoconductive antenna receiver. The near-field signal, corresponding to the electric field amplitude $|E_z|$ perpendicular to the substrate, was extracted by demodulating the higher harmonics nf (where $n \geq 3$) of the tip vibration frequency. A typical near-field amplitude image is obtained by integrating $|E_z|$ over the time domain, as shown in Fig. 4i."

The Figure R8 is added as Supplementary Fig. 13 in the Supplementary Information part.

Q5: Please explain the basic principle of Figure 4 (f-h).

R5: Figure 4f to 4h in our original manuscript refer to the numerical simulation results on spatial distributions of THz near-field amplitude (Figure R9c), charge density (Figure R9a), and Joule loss (Figure R9b) of an individual circular-shape β_{12} -Br nanosheet, which is placed onto a 500- μm -thick highly doped n-silicon substrate. Among them, the spatial distributions of THz near-field amplitude is for comparison with the experimental near-field measurement result shown in Figure R9d.

Figure R9. Simulated spatial distributions of charge density (a), Joule loss (b), and near-field intensity (c) within an individual β_{12} -Br nanosheet on a 500- μm -thick highly doped n -silicon substrate. The simulations are conducted in the frequency range of 0.5 – 2 THz. The diameters of the circular disks are all set at 5.64 μm . (d) Near-field image of an individual β_{12} -Br nanosheet on a 500- μm -thick highly doped n -silicon substrate.

We employed the FDTD (FDTD, Lumerical Solutions Inc.) method to calculate the near-field distribution, charge distribution, and Joule heat loss within an individual borophene circular nanodisk. According to the experimental measurement shown in Figure 4i in our original manuscript, the diameter of the disk is set as 5.64 μm . To simplify the calculation process and conserve computational resources, a perfect 2D circular disk model without a thickness parameter is used in our calculations. The nanosheet is placed on a 500- μm -thick highly doped n -silicon substrate, exactly as that used in the experimental near-field measurement. The surrounding environment is set as vacuum, with a refractive index of 1.0. The complex conductivity of borophene is described using the Drude model, with parameters adjusted to fit the experimental THz absorption spectra of the β_{12} -Br/PDMS composite thin films (see response to Q2 of Reviewer #1 above, and also Supplementary Note 2 in our revised manuscript). The THz near-field characteristics of a specific nanosheet can be readily calculated once its absorption cross-section spectrum is obtained.

To calculate the near-field distribution and compare it with the experimental near-field measurement, the scanning tip of the scattering-type scanning near-field optical microscope was modeled as a point dipole source raster-scanned across the simulation domain. The dipole, oriented perpendicular to the sample surface, was positioned at a fixed height of 200 nm above the substrate.

For each simulation, the perpendicular component of the electric field, $|E_z|$, was extracted in the frequency domain (0.5 to 2.0 THz) at a plane 100 nm above the surface and converted back to time-domain near-field optical maps via inverse Fourier transform. The $|E_z|$ was then integrated over the time domain to obtain the near-field distribution, as shown in Figure R8c above. For calculation of the charge density (ρ) and Joule loss (J) within the nanosheet, a broadband plane wave was employed to illuminate the nanosheet, whereby the electric field, \vec{E} , within the nanosheet was recorded. Afterwards, ρ and J can be obtained according to,

$$\rho = \varepsilon_0 \nabla \cdot \vec{E} \quad (\text{R13})$$

$$J = \frac{1}{2} \sigma \vec{E} \cdot \vec{E}^* \quad (\text{R14})$$

To ensure consistency with the experimental conditions, we employed the same broadband source (0.5 to 2.0 THz) in our simulations to excite an individual borophene nanosheet and performed spectral integration of the ρ and J over the same spectral range.

The calculated results are shown in Figure R9a–9c. The three figures visualized the dynamics of the free electrons upon the THz excitation within the circular borophene nanosheet. Specifically, the THz illumination induces strong electron oscillations within the nanosheet, which are subsequently reflected by the disk boundary (Figure R9a). This results in significant Joule loss occurring within the borophene nanosheets (Figure R9b), leading to attenuated electromagnetic fields inside the disk compared to the substrate (Figure R9c). This phenomenon is further corroborated by THz near-field optical measurements conducted on an individual β_{12} -Br nanosheet (Figure R9d). As a result, the Joule loss contributes to a remarkably high absorption efficiency of up to 43% for the nanosheet array at resonance, even for a monolayer thickness (Figure R3c). This highlights the efficient energy dissipation mechanism enabled by the rich free electrons within the β_{12} -Br nanosheets.

The simulation procedures of the $|E_z|$, ρ , and J are added in Supplementary Note 2 in the Supplementary Information part in our revised manuscript. In addition, a discussion explaining the basic principle of Figure 9a–9c, i.e., Figure 4f–4h, is added in paragraph 2 in page 13 as,

"...This results in significant Joule loss occurring within the borophene nanosheets (Fig. 4g), leading to attenuated electromagnetic fields inside the nanosheet compared to the substrate (Fig. 4h). This phenomenon is further corroborated by THz near-field optical measurements conducted on an individual β_{12} -Br nanosheet (Fig. 4i, see Method for numerical simulations). As a result, the Joule loss contributes to a remarkably high absorption efficiency of up to 43% for the nanosheet array at resonance, even for a monolayer thickness (Supplementary Fig. 9). This highlights the efficient energy dissipation mechanism enabled by the rich free electrons within the β_{12} -Br nanosheets...."